# Investigating ANN architectures and training to estimate SWE from snow depth

Konstantin F. F. Ntokas[1,2], Jean Odry[1], Marie-Amélie Boucher[1], and Camille Garnaud[3]

[1]Université de Sherbrooke, Department of Civil and Building Engineering, Sherbrooke, Canada
[2]Technische Universität Berlin, Berlin, Germany
[3]Environment and Climate Change Canada, Dorval, Canada

**Correspondence:** Konstantin Ntokas (konstantin.ntokas@yahoo.de)

**Abstract.** Canada's water cycle is driven mainly by snowmelt. Snow water equivalent (SWE) is the snow-related variable that is most commonly used in hydrology, as it expresses the total quantity of water (solid and liquid) stored in the snowpack. Measurements of SWE are, however, expensive and not continuously accessible in real time. This motivates a search for alternative ways of estimating SWE from measurements that are more widely available and continuous over time. SWE can be calculated by multiplying snow depth with the bulk density of the snowpack. Regression models proposed in the literature first estimate snow density and then calculate SWE. More recently, a novel approach to this problem has been developed and is based on an ensemble of multilayer perceptrons (MLPs). Although this approach compared favourably with existing regression models, snow density values at the lower and higher ends of the range remained inaccurate. Here, we improve upon this recent method for determining SWE from snow depth. We show the general applicability of the method through the use of a large data set of 234 779 snow depth-density-SWE records from 2878 non-uniformly distributed sites across Canada. These data cover almost four decades of snowfall. First, it is shown that the direct estimation of SWE produces better results than the estimation of snow density followed by the calculation of SWE. Second, testing several ANN structural characteristics improves estimates of SWE. Optimizing MLP parameters separately for each snow climate class gives a greater representation of the geophysical diversity of snow. Furthermore, the uncertainty of snow depth measurements are included for a more realistic estimation. A comparison with commonly used regression models reveals that the ensemble of MLPs proposed here leads to noticeably more accurate estimates of SWE. This study thus shows that delving deeper into artificial neural network theory helps improve SWE estimation.

## 1 Introduction

Snowmelt plays a major role in the hydrological cycle of many regions of the world. Casson et al. (2018) determined that snow accumulation and melt are the main drivers of the spring freshet in the Canadian subarctic region, and Pomeroy et al. (2011) demonstrated that over 80% of the annual runoff in the Canadian Prairies is derived from snowmelt. An accurate prediction of the accumulation and melting of snow is therefore of interest for various applications, including the management of reservoirs for hydroelectric power generation, irrigation and water supply, and climate impact studies.

Consequently, many hydrological models include a snow module to estimate the snow water equivalent (SWE). SWE is of great interest in hydrology because it describes the volume of water stored in the snowpack (e.g. Kirnbauer et al. (1994); Dozier (2011); Hock et al. (2006); Barnett et al. (2005); Seibert et al. (2014)). Kinar and Pomeroy (2015) reviewed techniques for measuring snowpack variables. The existing techniques for determining SWE are either time-intensive, due to manual snow surveys, or cost-intensive because of the necessary and expensive equipment. Sturm et al. (2010) estimated SWE measurements to be 20 times more expensive than snow depth measurements. SWE can also be calculated, however, using snow depth and the volumetric mass density of snow.

Snow depth can be measured inexpensively by ultrasonic distance sensors. Furthermore, light detection and ranging instruments (lidar) installed on aircraft can measure snow depth remotely (e.g. Painter et al., 2016; Kim et al., 2017). Lidar penetrates vegetation, which makes it a very good tool for measuring snow depth in forested environments. These observations also have a high spatial resolution, which makes lidar particularly interesting in mountainous areas where snow depth is highly spatially variable. At present, the high cost of using manned aircraft prevents this technique from becoming operational at a large scale, although Bühler et al. (2016) have demonstrated the potential for unmanned aerial surveys using low-cost, remotely piloted drones to measure snow depth at a large scale and at a high spatial resolution. Lettenmaier et al. (2015) discussed remote sensing techniques in hydrology and concluded that measurement of SWE from space "remains elusive". However, Environment and Climate Change Canada (ECCC) and the Canadian Space Agency (CSA) are currently collaborating on the Terrestrial Snow Mass Mission (Garnaud et al., 2019), a satellite mission that aims to measure SWE with a dual Ku-band radar.

A number of regression models have already been proposed to convert snow depth to SWE (e.g. Jonas et al., 2009; Sturm et al., 2010; Painter et al., 2016; Broxton et al., 2019). In general, these models first estimate snow density from snow depth and then calculate SWE. Sturm et al. (2010) argue that the variation of snow density is four times less than that of snow depth. Therefore, a model that measures the more dynamic parameter (snow depth) and estimates the more conservative parameter (snow density) offers promise. The models are trained on field measurements using regression analysis. Additional geophysical classifications are used to obtain more precise results. For example, Sturm et al. (2010) and Bormann et al. (2013) used the snow classification of Sturm et al. (2009) in which individual regression models were trained for each class. Another example of geophysical classification is given by Jonas et al. (2009). They applied an individual regression model to each month and elevation class to separate respectively the data both temporally and spatially.

Physics-based approaches have also been proposed for converting snow depth to SWE. For instance, Painter et al. (2016) used the snowmelt model *iSnobal*, which calculates snow density while incorporating snow ageing, mechanical compaction, and the impact of liquid water with adjustments for deposits of new snow. Input variables (e.g. incoming longwave radiation, soil temperature, net solar radiation), which are necessary for physical modelling, are not available in real time in Canada. Furthermore, physical-based models are, as Painter et al. (2016) mentioned, the logical choice for distributed SWE estimates. However, we aim for a conversion model based on data points which are sparsely scattered in time and space and that uses only variables available in real time. Further, physics-based models can be because of their complexity computationally expensive.

Recent studies have suggested the use of artificial neural networks (ANN) to estimate SWE. Snauffer et al. (2018) uses ANNs for multi-source data fusion over British-Columbia, Canada. SWE data from reanalysis products and manual snow survey are

used as network inputs and gridded SWE products are improved by the ANN. The following two studies uses ANNs to model the relationship between snow depth and snow density, which could then be used to obtain estimates of SWE. Broxton et al. (2019) applied an ANN model to a set of snow measurements at a very high spatial resolution. Their study focused on two approximately $100\,\text{km}^2$ areas in Arizona, USA, for both mid- and late-winter conditions, during which a total of 300 density-depth tuples were obtained manually. These tuples were used to train the neural network. Their model, which consisted of a simple network structure with one hidden layer containing 10 neurons, also incorporated other physiographic factors obtained from lidar measurements. The Levenberg–Marquardt algorithm optimized the model over a run of 50 epochs. The Broxton et al. (2019) approach allows for the use of high-resolution lidar measurements of snow depth; the ANN model then converts these measurements to snow density and subsequently to SWE to produce improved maps of depth, SWE, and snow density at a very high resolution.

Odry et al. (2020) also applied ANN to estimate snow density from snow depth, but they focused on developing a method that would be applicable over a very large spatial extent. In their study, they used almost $40\,000$ measurements from approximately $400$ non-uniformly distributed sites across the province of Quebec, Canada. This study covered a period of 45 years. In contrast to Broxton et al. (2019), the available snow measurements were spatially distant and temporally irregular. The ANN incorporated meteorological data as additional explanatory variables to support the estimation of density. Odry et al. (2020) used an ensemble of multilayer perceptrons (MLPs, a type of ANN) to provide estimates, at least in part, of the uncertainty associated with converting snow depth to density. A comparison of the ANN model of Odry et al. (2020) with the regression models of Jonas et al. (2009) and Sturm et al. (2010) in a leave-one-out setup showed that the ANN model provided the most accurate results. However, Odry et al. (2020) also noted that all three models performed relatively poorly for very low or very high snow densities. Given that high densities generally correspond to the beginning of the melt period, it is particularly crucial to obtain accurate SWE values for this period of the year.

In this study, we build a model to estimate SWE from in-situ snow depth measurements and several indicators derived from gridded meteorological time series. This study is a follow-up to the work of Odry et al. (2020). Odry et al. (2020) showed that ANN ensembles are a good method for a snow depth–SWE conversion model, but did little work on the optimization of the architecture. Furthermore, we suspect that ANNs are capable of estimating SWE being the direct output of the ANN, because in theory ANN are capable of representing any continuous function, which is discussed in Sect. 2.1. Therefore, we will mainly test two hypotheses: (1) using SWE instead of density as the target variable for the ANN produces more accurate estimates of SWE; (2) testing several options of ANN structural characteristics (e.g. optimization algorithm, activation function, parameter initialization, increasing the number of parameters) improves estimates of SWE. So far, also no input uncertainty is considered by Odry et al. (2020), which leads us to a small test of input uncertainty on one input variable, snow depth. The input variables section was determined by the Spearman's correlation in Odry et al. (2020), indicating the monotonic relationship. In this study we determine the input variables directly on the network (expect for pre-filtering in Sec. 3.2), because the monotonic relationship as the only criteria can be misleading in representing the complexity of ANNs. When looking at the snow classification scheme introduced by Sturm et al. (2009), Odry et al. (2020) used a data set which contains two snow classes. In this study we apply the proposed snow depth–SWE conversion framework to an extended data set scattered sparsely

and non-uniformly over the entire area of Canada, which tests its applicability to all proposed snow class zones except of ice.

We will build two ANN models. One which takes the same architecture for the entire data set and one which uses an individual architecture for each snow class to give a greater representation of the geophysical diversity of snow.

The remainder of the paper is organized as follows. In Sect. 2, we present the main background elements and the essential literature review. This review includes the mathematical theory of MLPs, the applied snow climate classes (Sturm et al., 2009), the statistical regression models proposed by Sturm et al. (2010) and Jonas et al. (2009) which are used for comparison against

our neural network models, and the performance assessment metrics. Section 3 presents the experimental protocol, including the available data and input variables and the procedure for determining the MLP architecture. Results are presented in Sect. 4, and our conclusions are summarized in Sect. 5.

## 2 Literature review

### 2.1 The multilayer perceptron as a basic tool

Following Goodfellow et al. (2016), a multilayer perceptron (MLP) is a fully connected, feed-forward artificial neural network, meaning that information can only travel in one direction within the network. Being fully connected entails that no initial assumptions need to be made in regard to the data.

The goal is to approximate a desired function $f$ such that $y = f(\boldsymbol{\theta}, \boldsymbol{x})$, where $y$ is the target variable (in our case SWE), $\boldsymbol{x}$ represents the input variables, and $\boldsymbol{\theta}$ represents some parameters, namely weights and biases. The use of non-linear activation

functions in each neuron enables the network to approximate non-linear functions $f$.

To determine the parameters, one can perform an optimization over the space of the weights and biases by using a training data set. To do so, the parameters are initialized commonly at random and close to zero by following either a uniform or Gaussian distribution. For regression problems, the mean square error (MSE) is commonly used as the objective function in the optimization. Goodfellow et al. (2016) show a derivation of the MSE from the maximum likelihood estimator. During

training, the training data set is presented multiple times to the model, and the term *epoch* is used to denote one run over the entire training data set. It is recommended that between each epoch the order of the data points of the training set is permuted. This prevents the model from learning features in the order of insertion. The training of a MLP is related to an ordinary optimization. The difference is that we optimize directly for a training data set with the goal that the measure of performance of a validation data set is also optimized. Commonly, there will be a turning point where the training error continues to

decrease, but the validation error starts to increase. For a single deterministic MLP, this turning point would be chosen as the best number of epochs for the model. When training an ensemble of MLP, however, it is ideal to maintain diversity among the ensemble members to cover the range of uncertainty pertaining to the target variable. If all members of the ensemble were to be trained until the aforementioned turning point, they would all become very similar, and it would become pointless to use an ensemble in the first place. For this reason, we adopt here the protocol proposed by Boucher et al. (2010) and monitor rather

the performance of the entire ensemble at each epoch (see Sect. 4.1).

During the optimization, a non-convex function is solved; this function contains multiple minima having a similar performance. The objective function with respect to the parameters is non-convex because of several symmetric configurations of a neural network. Thus, exchanging the associated bias and weights of one neuron with another neuron in the same layer entails the same results. Furthermore, ANN applications usually use input variables that are related to each other. The interchangeability of dependent input variables results in multiple parameter sets having a similar performance. The number of dimensions in the optimization is equal to the number of parameters, and it is also argued by Goodfellow et al. (2016) that local minima are rare in high-dimensional spaces. Dauphin et al. (2014) point out that with increasing dimensions, the ratio of the number of saddle points to the number of local minima increases exponentially. Therefore, an optimization method is needed that does not become stuck in saddle points.

The concept of stochastic gradient methods has been introduced to avoid saddle points. These methods are based on the ordinary steepest gradient method; however, rather than using the entire training data set (batch) at once, a number of data records (or "minibatches") are taken for each iteration during the optimization. Consequently, the optimization surface is slightly altered at each iteration. This can even sometimes help to escape shallow local minima. For the interested reader, Goodfellow et al. (2016) provide an overview of various optimization algorithms.

The related studies of Broxton et al. (2019) and Odry et al. (2020) both used the second-order Levenberg–Marquardt algorithm. Broxton et al. (2019) worked with the *Matlab deep learning toolbox* and Odry et al. (2020) used an in-house built *Matlab* toolbox by Dr Kris Villez, derived from the *Matlab deep learning toolbox*. The conversion of the codes into *Python* revealed that the Levenberg–Marquardt algorithm has an oscillating convergence of the training error because the minimum is over-jumped multiple times. This results in a higher computational cost. The *Matlab deep learning toolbox* only shows successful parameter updates where the error of the objective function is decreased. This conceals multiple failures. Further discussion about the drawbacks of the *Matlab toolbox* is available in Kwak et al. (2011). Furthermore, the dimensions of optimization increase exponentially with the size of a neural network. Therefore, we will only consider first-order optimizers, as these are computationally more efficient. Goodfellow et al. (2016), in their overview of optimization methods, conclude that stochastic gradient methods with adaptive learning rates are the best choice for a first-order optimizer. Schaul et al. (2014) compare stochastic gradient methods by testing them on small-scale problems and found out that the algorithms RMSProp and AdaDelta produce good results. The drawback of RMSProp is the need for a global learning rate, which must be defined by the user. To address this problem, Zeiler (2012) introduced the algorithm AdaDelta, which eliminates the requirement of this global learning rate.

There are several reasons that support the use of an ensemble rather than a single model. First, random parameter initialization can end up in different local minima on the parameter surface with similar performance. This situation is related to the concept of equifinality, introduced by Bertalanffy (1968) for open systems stemming from the work of the biologist Hans Driesch. Therefore, using an ensemble accounts for the uncertainty of the model parameters. Second, the ensemble offers the possibility of probabilistic simulations. Therefore, probabilistic evaluation methods, introduced in Sect. 2.4, can be used. This leads to greater insight into model performance. Third, the different members of the ensemble can be used in a Kalman filter or a particle filter for data assimilation in a hydrological model.

Finally, in regard to the architecture of the MLP, through the universal approximation theorem, it is proven that any continuous function can be approximated by a feed-forward neural network with a single hidden layer under mild assumptions. This theorem was first proven by Cybenko (1989) for the sigmoid activation function. Hornik (1991) showed the same theorem is independent of the choice of the activation function but assumes that the output layer is linear, which is the case for regression problems. The theorem does not make any claims about guidelines of the architecture or about the learning ability of the model. In ANN applications, therefore, the perfect approximation of the function is not obtained. Rather, there is a trade-off between computational cost, the time to test different architecture compositions, and accurate approximation.

According to Goodfellow et al. (2016), the most common activation functions are the hyperbolic tangent (tanh) and the rectified linear unit (ReLU) functions. The calculation of the derivative of tanh is computationally expensive, and the derivative of the tanh function for large values is almost zero, which slows down learning. The derivative of the ReLU function is easy to compute, and the derivative stays constant for large positive values. However, the derivative is zero for negative values, which stops the training within the optimization. This phenomenon is known as "dying neurons". The Leaky ReLU function can tackle this issue by assigning a slightly positive slope, typically $0.01$, to the negative part of the ReLU function.

## 2.2 Snow classification

In this section, we introduce a snow classification scheme proposed by Sturm et al. (2009). These snow classes are used to train a different ensemble of MLP for each snow class. In accordance with the findings of Sturm et al. (2010) and Bormann et al. (2013), we expect that this approach will result in an improved accuracy for estimates of SWE compared to training an ensemble over the data for Canada as a whole.

The snow classification system separates the world map into seven different snow classifications at a resolution of $0.5° \times 0.5°$. Snow can vary in character (e.g. snow found in areas close to coasts consists mainly of wet snow, whereas snow located in the continental interior is characterized by dry snow). Therefore, the data within one snow class should be systematically more consistent and may show less variability than taking the snow data as a whole. This can help in our study, as the MLP must capture fewer features, which may lead to a better optimization and, in turn, improved performance. The snow classes across Canada are presented in Fig. 1.

Some sites close to the coast are classified as *water*, because the resolution of the snow classification map is coarse. Furthermore, the snow class *ice* only contains 124 records. For these two classes, we assign them to the nearest—in terms of physical distance—accepted snow class.

The empirical cumulative distribution function (ECDF) of snow depth, SWE, and snow density for each snow class is depicted in Fig. 2. The number of records in each snow class is presented in Fig. 2b. The *ephemeral snow* class is found only on Vancouver Island, which also explains the few records of this class. Furthermore, the high annual precipitation and marked variability in elevation across Vancouver Island explains the very high variability of snow depth, SWE, and snow density for this class. When comparing the *taiga* and *tundra snow* classes, one can observe that snow depth is lower in the *tundra snow* class. Sturm et al. (1995) point out that *tundra* is characterized by a thin snow layer consisting mainly of wind slabs and depth hoar. Nonetheless, the density distribution of the *tundra snow* class is higher; this may reflect the wind densification effect

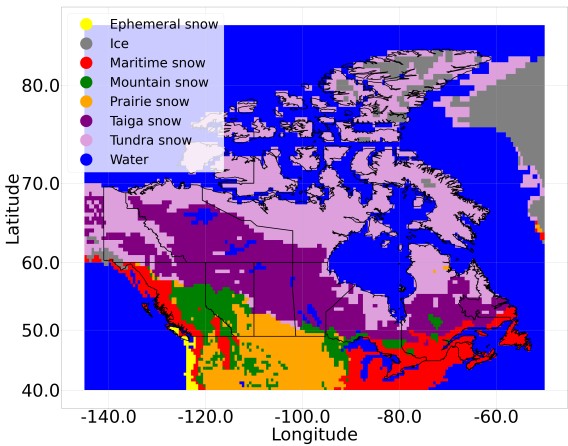

**Figure 1.** Snow classes across Canada as defined by Sturm et al. (2009)

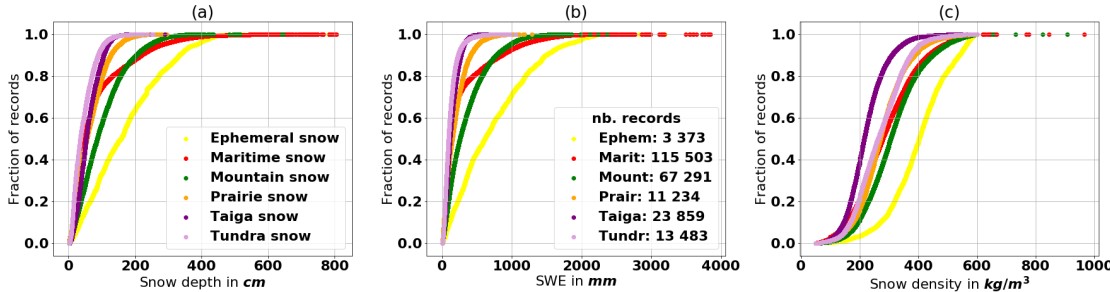

**Figure 2.** The empirical cumulative distribution function (ECDF) of the variables snow depth, SWE, and density for each snow class

and compaction of snow during a longer duration in this region. The *maritime*, *mountain*, and *ephemeral snow* classes have a higher density distribution than the *prairie* and *taiga snow* classes because of the greater oceanic influence on the *maritime* and *ephemeral* regions. This influence results in wetter snow and, therefore, higher snow densities within these latter regions. Most records of the *mountain snow* class are found in the Rocky Mountains along the border of British Columbia and Alberta, an area influenced by the Pacific Ocean, thereby producing a higher density snow. A more thorough analysis of the data revealed that the high number of records in the *maritime* and *mountain* snow classes is due to records from snow pillows in British Columbia, collected between 1996 and 2011. Snow pillows continuously measure the weight of the snow, which is then presented as a daily measurement for some sites in this data set. British Columbia contains almost half of all snow data records for Canada.

## 2.3 Regression models

In this section we introduce two regression models from the literature, which we use as benchmarks to compare with our MLP-based conversion model, using the performance indicators described in Sect. 2.4.

### 2.3.1 The Sturm model

Sturm et al. (2010) proposed a model to convert snow depth into bulk density for the snow cover. The estimated density is then used to calculate SWE. As observed by Dickinson and Whiteley (1972), the range of snow density is smaller than the range of snow depth. Therefore, Sturm et al. (2010) claim that estimating the snow density is probably the most accurate way to estimate SWE. The regression model is formulated as:

$$\rho_{sim} = (\rho_{max} - \rho_0)[1 - e^{-k_1 SD_{obs} - k_2 DOY_{obs}}] + \rho_0, \tag{1}$$

where $[\rho_{max}, \rho_0, k_1, k_2]$ are parameters, $SD_{obs}$ is the snow depth, and $DOY_{obs}$ is the number of days since 01 September. The value of $DOY_{obs}$ is 0 on 01 January and therefore, corresponds to -122 for 01 September until 243 for 30 August. A different regression model is run for each snow class. The snow classes are defined by Sturm et al. (2009). The parameters for each snow class are obtained by carrying out an optimization on a training data set associated with that snow class. The RMSE between the estimated and the measured snow density is used as an objective function in the optimization.

### 2.3.2 The Jonas model

Jonas et al. (2009) propose a simple linear regression model that uses elevation classes to separate the data spatially and divide the data into months. This model also estimates density, and from this density, it calculates the SWE. The regression is defined as:

$$\rho_{sim} = a\, SD_{obs} + b + offset_{reg}, \tag{2}$$

where $a$ and $b$ are the parameters of the linear regression, $SD_{obs}$ is the observed snow depth, and $offset_{reg}$ is a regional specific parameter. The data set is split by month and into three elevation classes, denoted by $x < 1400\,$m, $1400\,$m$\leq x < 2000\,$m, and $x \geq 2000\,$m, where $x$ is the elevation of the site. After that, the parameters $a$ and $b$ are derived by fitting a linear regression to each portion of the data set. This process produces 36 independent linear regression models. After solving for the parameters of the linear regressions, we perform a simulation without $offset_{reg}$. The regional specific parameter $offset_{reg}$ is thus the average of the model residual between the simulated and observed density of samples in a given region. In our study, the snow classes defined by Sturm et al. (2009) are used as these regions. Note that the separation into the snow classes for the $offset_{reg}$ calculation is applied independently after the linear regressions are performed for each month and elevation class. The $offset_{reg}$ parameter eliminates the regional bias.

## 2.4 Model evaluation

In this section, we introduce deterministic evaluation metrics (Sec. 2.4.1) followed by metrics that can be applied onto a probabilistic ensemble simulation (Sec. 2.4.2 – 2.4.5). Gneiting and Raftery (2007) evaluate the quality of ensemble simulation by scoring rules. A scoring rule quantifies the quality on the basis of the predicted distribution of the ensemble and the observation. A strictly proper scoring rule is defined such that it has a unique global minimum, which is the distribution of the

observation itself. This minimum shows the highest possible performance of the simulation. The probability density function (pdf) is usually derived by distribution fitting to the members of the ensemble or by simply taking the empirical cumulative distribution function (cdf) over the members and deriving the pdf from this. In this study, only the empirical case is used to eliminate the uncertainty added by the distribution fitting.

Furthermore, an ensemble is said to be reliable if the relative frequency of the event, for a given simulation probability, is equal to the simulation probability.

### 2.4.1   Deterministic evaluation metrics

Deterministic evaluation metrics quantify the performance based on the single outcome of a deterministic model and the observation. Here, the median of the ensemble is considered as a deterministic simulation on which these metrics can be

applied. The most popular measures are the Mean Absolute Error, the Root Mean Square Error and the Mean Bias Error.

    The Mean Absolute Error (MAE), Root Mean Square Error (RMSE) and Mean Bias Error (MBE) are defined as

$$MAE = \frac{1}{n} \sum_{i=1}^{n} |y_{sim_i} - y_{obs_i}| \tag{3}$$

$$RMSE = \sqrt{\frac{1}{n} \sum_{i=1}^{n} (y_{sim_i} - y_{obs_i})^2} \tag{4}$$

$$MBE = \frac{1}{n} \sum_{i=1}^{n} y_{sim_i} - y_{obs_i} \tag{5}$$

In Eq. 3–5, $n$ is the number of records, $y_{sim_i}$ is the simulated output of the model and $y_{obs_i}$ the observation, both associated with the $i^{th}$ record. Note that the RMSE is similar to the MAE, but gives records with large errors a greater weight and therefore, penalizes outliers more.

### 2.4.2   The ignorance score

Roulston and Smith (2002) introduced the ignorance score, which is defined as:

$IGS(f_M, o) = -log_2(f_M(o)),$                      (6)

where $f_M$ is the simulated pdf, and $o \in \mathbb{R}$ is the observation. Since the values of a pdf are between 0 and 1, the ignorance score takes values greater or equal 0. Furthermore, it is not defined for $f_M(o) = 0$. It assigns the minimum value of zero if the probabilistic forecast predicts the observation with a probability equal to 1, indicating a perfect simulation. Therefore, the ignorance score is a strictly proper scoring rule. To assess the model, the average of the ignorance scores over all $n$ records is

taken, defined as:

$$\overline{IGS} = \frac{1}{n} \sum_{j=1}^{n} -log_2(f_{Mj}(o_j)), \tag{7}$$

where $f_{M_j}$ is the pdf derived from the ensemble of the $j^{th}$ simulation, and $o_j$ is the $j^{th}$ observation.

In the empirical case, i.e. without distribution fitting, we consider the sorted predicted ensemble with $m$ members by $\{s_1, s_2, ..., s_m\}$. The cumulative distribution function is simply the staircase function over the members. From this, we construct a pdf by assigning each area between two members:

$$f_M(x) = \frac{1}{m(s_i - s_{i-1})} \quad \forall x \in [s_{i-1}, s_i].$$

If the observation coincides with a member, the larger probability of the two adjacent areas is taken. Furthermore, $f_M(x)$ is set to $0.001$, if the observation lies outside of the ensemble.

### 2.4.3 The continuous ranked probability score

Given that the ignorance score evaluates the simulated probability function only at the point of observation, no information about the area surrounding the observation or the shape of the probability function is included. The continuous ranked probability score (CRPS) addresses this drawback by working directly on the cdf. The CRPS of one record is defined as:

$$CRPS(F_M, o) = \int_{-\infty}^{\infty} (F_M(w) - \mathbf{1}_{[o,\infty)}(w))^2 dw, \tag{8}$$

where $F_M$ is the cdf derived from the ensemble, and $o \in \mathbb{R}$ is the observation. The integrand is the squared difference between the cdf derived by the ensemble and the cdf derived by the observation. The CRPS takes values in $\mathbb{R}_0^+$, where zero is assigned if the predicted cdf is equal to the cdf derived by the observation. This is the perfect simulation. This describes the CRPS as a strictly proper scoring rule. When calculating the empirical CRPS, the staircase function over the members within the ensemble is taken as the cdf.

Hersbach (2000) presents a decomposition of the empirical CRPS into a reliability part and the potential CRPS. We denote the ensemble of one record by $\{x_1, ..., x_m\}$. The reliability part is then defined as:

$$Reli = \sum_{i=0}^{m} g_i (o_i - p_i)^2, \tag{9}$$

where $o_i$ is the relative frequency of the observation being smaller than the midpoint of the bin $[x_i, x_{i+1}]$, $p_i$ is $\frac{i}{m}$, the value of the cdf of the ensemble at $x_i$, and $g_i$ is the average bin size of $[x_i, x_{i+1}]$. To get the reliability score of the model, the average over all records is taken.

Small values of the reliability portion testify to the high reliability of the ensemble simulation. The reliability portion is highly related to the rank histogram, introduced in the next section (Sect. 2.4.4). The difference is that the reliability part takes the spread of the ensemble into account, which is not incorporated into the rank histogram. When the reliability part is subtracted from the empirical CRPS, the potential CRPS remains. The potential CRPS is the same as the CRPS when the reliability part is equal to zero, which indicates a perfectly reliable model. The potential CRPS is related to the average spread of the ensemble. A narrow ensemble leads to a small potential CRPS.

### 2.4.4 The rank histogram

The rank histogram was developed independently and almost simultaneously by Anderson (1996), Hamill and Colucci (1997), and Talagrand et al. (1997). It determines the rank of each observation within the associated predicted ensemble. The ranks of all observations are then presented in a histogram. A perfectly reliable ensemble forecast shows a flat rank histogram. Furthermore, the rank histogram can reveal information about bias and under- and overfitting. For instance, if the rank histogram shows a decreasing trend with higher ranks, the model has a positive bias. An increasing trend with higher ranks shows a negative bias. Furthermore, a u-shaped rank histogram indicates a too-small ensemble spread. This limited spread is related to an overconfident model, which means that the model ignores some parts of the uncertainty. Several reasons explain this, including the fit of the model. If a model is trained too long, it is trained too specifically on the training data. Consequently, the model performs well on the training data, but it lacks generality, meaning that the model is incapable of simulating an unseen data set. In contrast, a bell shape indicates a too-large ensemble spread. Note that a flat histogram is a necessary condition for a model to be reliable, but this is not a sufficient condition. Examples and further discussion are given by Hamill (2001).

### 2.4.5 The reliability diagram

The reliability diagram is a visual tool for determining the reliability of an ensemble forecast. The relative frequency of the observation, given the probability of the ensemble, is plotted against the probability of the ensemble.

To construct a reliability diagram, we work with the empirical staircase function over the ensemble members as cdf. Let us assume that we have $k$ bins, denoted by $b_1, ..., b_k$. These bins represent a probability and, therefore, their sizes are $bs_1, ..., bs_k \in (0, 1]$. Furthermore, all bins are centred on $0.5$, therefore indicating an interval around the median. Thus, a bin $b_i$ is indicated by the interval $[s_i, e_i]$. We find the $s_i$- and $e_i$-quantiles within the ensemble distribution, denoted by $s_{en_i}$ and $e_{en_i}$, respectively. Interpolation is used if $s_i$ and $e_i$ lie between the steps. Therefore, the interval $[s_{en_i}, e_{en_i}]$ shows the interval around the median of the ensemble distribution, which has the probability equal to $bs_i$. Then, we determine the relative frequency of the observations found within this interval, denoted by $o_i$. The points $(bs_i, o_i)$ are plotted against the line of the identity function. If all points fall on that line, the model is perfectly reliable, meaning that the relative frequency of the observation is equal to the simulated frequency.

### 2.4.6 Skill scores

Skill scores enable comparing simulated model outputs of different magnitudes. The skill score $SS$ is defined as:

$$SS = \frac{score_s - score_{ref}}{score_{perf} - score_{ref}}, \tag{10}$$

where $score_s$ is the score of the actual simulation, $score_{ref}$ is the score of a reference simulation, and $score_{perf}$ is the score of a perfect simulation. Climatology or persistence is often used as a reference simulation. A skill score takes values from $(-\infty, 1]$. A value of $1$ indicates that the actual simulation is a perfect simulation. A negative skill score indicates that the actual

simulation is less accurate than the reference simulation.

A variation of climatology is taken as a reference simulation, because SWE measurements are not continuous over time in the data set. Records from the training and validation data sets at the same location and within a time window of $\pm15$ days around the date of the corresponding observation are used to build up a reference ensemble having 20 members. For 77 % of the testing records, an ensemble from climatology could be obtained in this manner. In the calculation of the skill score in Eq. (10), the mean over the individual scores is taken within the fraction. This enables us to use only the portion of records where we found the 20 members to calculate the reference scores.

### 2.4.7 Sensitivity score

Following Olden and Jackson (2002), we set input-output records as references where the output is the 20th, 40th, 60th, and 80th percentiles of the target variable SWE in the validation data set. We take the average of 1000 records around the percentile to maintain the generality of the reference record, because single records can deviate from normality. Subsequently, one input variable is perturbed at a time by taking uniformly distributed values between the maximum and minimum values in the validation set of the considered input variable. The perturbed inputs are inserted into the network, and the perturbed output is generated. The sensitivity score is then calculated by

$$SensS = \frac{1}{4}\sum_{j=1}^{4}\frac{1}{20}\sum_{m=1}^{20}\sqrt{\frac{1}{n}\sum_{i=1}^{n}\left(y_{pert_{im}} - y_{ref_{jm}}\right)^2} \tag{11}$$

where is $y_{pert_{im}}$ is the perturbed output of record i and ensemble member m and $y_{ref_{jm}}$ is the output of the reference percentile j and ensemble member m. Thus, the sensitivity score is the RMSE between the perturbed and reference output, averaged over the four reference percentiles and all members. Therefore, it measures the change in the output of the trained ensemble network associated with the change in one input variable.

## 3  Experimental protocol

### 3.1  Data availability

Environment and Climate Change Canada (ECCC) through Brown et al. (2019) and Ministère de l'Environnement et de la Lutte contre les Changements Climatiques (MELCC) provided a Canada-wide snow data set, which includes SWE, snow depth, and snow density. For the remainder of this paper, this data set will be denoted as the Canadian historical snow survey (CHSS).

For this study, we use the above-mentioned CHSS snow data, collected from 01 January 1980 to 16 March 2017. It consists of 234 779 measurements from 2878 sites. Figure 3 presents some characteristics of the CHSS data set. The distributions of SWE and SD present a right-skewed gamma distribution. Snow density is almost normally distributed in a range from $50\,\mathrm{kg\,m^{-3}}$ to $600\,\mathrm{kg\,m^{-3}}$ with a few outliers at higher values. This distribution is related to the retrieval of two different data sets. The data

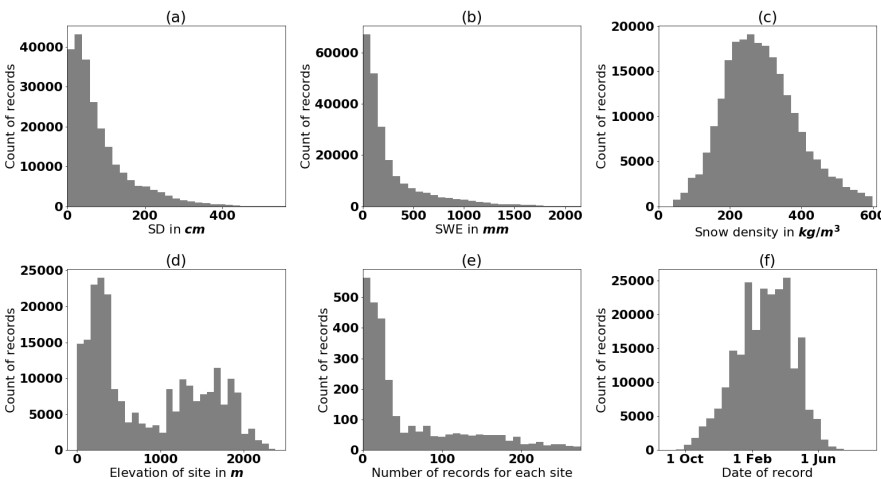

**Figure 3.** Distribution of the records in the Canadian historical snow survey (CHSS) data set for (a) snow depth, (b) SWE, (c) snow density, (d) elevation and (f) the date of the record; The year starts on 01 September to cover the Northern Hemisphere winter season over two calendar years; (e) distribution of number of records for each site; that upper outliers (0.1%) are excluded in (a), (b) and (c); furthermore, the upper 2.5% of stations are not shown in (e); the maximum in (e) is at 3203 and the mean at about 82 records for each station;

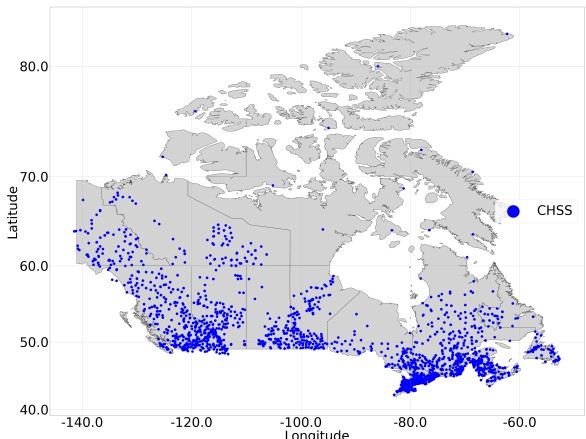

**Figure 4.** Location of the sites of the Canadian historical snow survey (CHSS) data set

from ECCC is bounded by the interval $[50, 600]$ for snow density, reflecting snow having a density of up to $600 \, \mathrm{kg} \, \mathrm{m}^{-3}$. Above this threshold, snow begins to transform into ice. The higher densities encountered in the MELCC data are due to the presence of an ice layer in the snow pack or to measurement artifacts. Note that in Figure 3a – c, the upper 0.1% outliers are excluded from the plot. Figure 3e shows a histogram of the number of records for each site, where the upper 2.5% stations are excluded from the plot. The mean is about 82 records per site and the maximum is 3203. This shows the temporal sparsity of the data set. Figure 4 shows the location of sites within the CHSS data set.

We retrieved daily total precipitation, snow density, as well as the maximum and minimum temperatures for each site of the CHSS from the ERA5 atmospheric reanalyses of the European Centre for Medium-Range Weather Forecasts (ECMWF) provided by the Copernicus Climate Change Service (C3S). We apply a temperature correction, dependent on elevation, to the minimum and maximum temperatures, because the reanalysis grid has a relatively low spatial resolution. The modified temperature $t_{mod}$ is defined as:

$$t_{mod} = t_{orig} + lapserate \, (elev_{site} - elev_{grid}),$$ (12)

where $t_{orig}$ is the original temperature of ERA5, $elev_{site}$ and $elev_{grid}$ are the elevation of the site and the grid point of ERA5 reanalysis, respectively. We apply a constant lapse rate of $-6 \, °C \, km^{-1}$, a value consistent with studies in the Rocky Mountains (e.g. Dodson and Marks, 1997; Bernier et al., 2011) and the global mean environmental lapse rate of $-6.5 \, °C \, km^{-1}$ (Barry and Chorley, 1987).

## 3.2 Explanatory variables

From snow depth, snow density, total precipitation, and temperature, we obtain the following explanatory variables. This initial pool of variables is based on Odry et al. (2020), except for snow density from ERA5. Snow density is included to test its influence on the simulation. Snow density is not available in real-time and thus, cannot be used in operation.

- averaged daily snow density (ERA5),

- snow depth,

- number of days since the beginning of winter,

- number of days without snow since the beginning of winter,

- number of freeze-thaw cycles; threshold for freezing and thawing is set at $-1 \, °C$ of the maximum and at $1 \, °C$ of the minimum temperature, respectively

- the degree-day index, i.e. accumulation of positive daily temperatures since the beginning of the winter,

- the snow pack aging index, i.e. the mean number of days since the last snowfall weighted by the total solid precipitation on the day of the snowfall,

- the number of layers in the snowpack estimated from the timeline and intensity of solid precipitation. A new layer is considered to be created if there is a three-day gap since the last snowfall,

- accumulated solid precipitation since the beginning of the winter,

- accumulated solid precipitation during the last $n$ days,

- accumulated total precipitation during the last $n$ days,

- mean average temperature during the last $n$ days (average temperature is taken as the mean of the maximum and minimum temperatures).

**Table 1.** Variables having the largest absolute Spearman correlation between the target variable and the last three explanatory variables for $n$ ranging between 1 to 10 days

| Variable | SWE | Snow density |
| --- | --- | --- |
| Accum. solid precipitation in the last | 10 days | 3 days |
| Accum. total precipitation in the last | 10 days | 10 days |
| Mean average temperature in the last | 6 days | 7 days |

We set the beginning of winter as 01 September because the seasonal distribution of the CHSS data set has the first snow records starting from mid-September, with the exception of some outliers. The separation of precipitation into solid and liquid parts is done by:

$$p(t_{av}) = \frac{1}{(1 + e^{-1.54 + 1.24 t_{av}})}, \tag{13}$$

where $p(t_{av})$ is the probability of snow, dependent on the average temperature $t_{av}$. Jennings et al. (2018) showed by using precipitation data from the Northern Hemisphere that this logistic regression model outperforms any temperature threshold separation model. To determine how many days are considered in the last three explanatory variables, we calculate the Spearman correlation between the target variable SWE or snow density and the range of 1 to 10 days. The results are presented in Table 1. The results in this table, along with the other above-mentioned variables, are used as input variables for the network.

Note that for accumulated solid and total precipitation the strongest correlation can be found at the upper limit for target variable SWE. However, we want to include the effect of short term variables and consider only a range of up to 10 days for the short term variables. Further discussion is given in the conclusion (Sec. 5).

Furthermore, we want to test the incorporation of input uncertainty on one variable, namely snow depth. According to the WMO (2014), the error of snow depth measurements should not exceed $\pm 1\,\text{cm}$ if the snow depth is less than $20\,\text{cm}$ and $\pm 5\,\%$

if the snow depth is greater or equal to $20\,\text{cm}$. To give an equal weight to the uncertainty of inputs and parameters, we perturb each record of the training data set 20 times with the above-mentioned variability. The model is then trained on the perturbed data.

### 3.3 Tested characteristics

The data set is divided randomly into three parts: training, validation, and testing sets (Hastie et al., 2009), each having a
405 proportion of one-third. The training set is used for training the MLP ensembles. The validation set is used to optimize the architecture of the network. The results are presented in Sec. 4.1. The testing set is used for the final model evaluation on an unseen data set and to compare with the regression models in Sect. 4.2 and 4.3, respectively. The column *Reference* in Table 2 shows the initial setup of a single MLP. This setup is used for all 20 members in the ensemble. Note that we always track the performance of the model on the validation data set over a range of epochs until evaluation metrics show worse results,

indicating overfitting of the model. This determines the correct training time. The number of neurons in the hidden layer for the reference setup is derived from the proposed rules of thumb of Heaton (2008) and Hecht-Nielsen (1989). The number of members is set at 20 because this number showed consistent results during several trials. The characteristics shown in Table 2 (first six rows in *Options*), are tested one at a time according to the *ceteris paribus* principle. Subsequently, characteristics that show improvements are tested in combination. Further testing of other combinations can reveal correlations between

characteristics, but will not be covered in this study. The results of these tests are shown in Sec. 4.1.1. To determine the selection of input variables, we carry out a sensitivity analysis of their importance. This refers to row seven in Table 2. The sensitivity score, introduced in Sect. 2.4.7, is used to determine the order of the importance of input variables. Subsequently, a stepwise reduction of input variables, starting with the least important variable, is performed. The results are shown in Sect. 4.1.2. Two models are then built. One model uses the data set of the whole territory to train on and consists of one ensemble

of MLPs. For the other model, the data set is split into snow classes and this results in one individual ensemble for each snow class. Thus, in the MMLP model the snow class is determined and the associated MLP ensemble is taken. This returns one ensemble for one set of input variables, as in the single MLP ensemble model. We will refer to the the first model as single MLP ensemble model (SMLP) and to the latter as multiple MLP ensembles model (MMPL). For each model, we perform a cross-analysis over the number of epochs and the number of hidden neurons. We showed in Sec. 2.2 that different snow classes

show different variability of snow records. The MMLP model tries to capture this through ANN ensembles with different sizes, because adding neurons to the hidden layer increases the ability to approximate more complex snow pattern. The number of parameters (the weights and biases) of the model are thus increased. This complicates the optimization and entails a longer training time to obtain the desired parameters. The results are shown in Sec. 4.1.3 and 4.1.4 for the cross-analysis and the final structure of the SMLP and MMLP model, respectively.

**4   Results**

The results are divided into three sections. In Sect. 4.1, we discuss the determination of the MLP ensembles' architecture, following mainly the outline of Sect. 3.3. Section 4.2 presents the performance of the final SMLP and MMLP models on the testing data set. In Sect. 4.3, we compare the final SMLP and MMLP models and the regression models introduced in Sect. 2.3.

**4.1   Results for tested characteristics**

First in Sec. 4.1.1, we discuss the results of the more general architecture characteristics of the MLPs, referring to the first six rows in Table 2. Sec. 4.1.2 contains the discussion on input variable selection. In Sec. 4.1.3, we discuss the results used for the determination of the number of neurons and number of epoch, for SMLP and MMLP models individually. The final setup of both models are presented in Sec. 4.1.4

**Table 2.** A summary of the reference MLP ensemble setup and the tests performed to obtain the final MLP architecture. The parameter initialization [1] is given by Goodfellow et al. (2016), where $m$ indicates the number of inputs, the parameter initialization [2] is given by Glorot and Bengio (2010), where $m$ and $n$ indicate the number of inputs and outputs, respectively. The parameter initialization [3] is our suggestion; SMLP refers to the single MLP ensemble model and MMLP to multiple MLP ensembles model; note that we track the performance of the model on the validation data set over a range of epochs until evaluation metrics show worse results to determine the correct training time;

| Tested for | Characteristic | Reference | Options | | |
|---|---|---|---|---|---|
| SMLP | Target variable | SWE | Density | | |
| | Input uncertainty of snow depth | No | Yes | | |
| | Activation function in the hidden layer | tanh | ReLU | Leaky ReLU | |
| | Optimization algorithm | AdaDelta | RMSProp | | |
| | Parameter initialization | $U(-1,1)$ | $\sim U\left(-\frac{1}{\sqrt{m}}, \frac{1}{\sqrt{m}}\right)^1$ | $\sim U\left(-\sqrt{\frac{6}{m+n}}, \sqrt{\frac{6}{m+n}}\right)^2$ | $\sim U(-2,2)^3$ |
| | Shuffling data before each epoch | No | Yes | | |
| SMLP | Input variables | 12 | 1-12 | | |
| SMLP and MMLP | Number of hidden neurons | 10 | 2-200 | | |
| | Number of epochs | suitable range | suitable range | | |
| not tested | Batch size | 100 | | | |
| | Number of members in the ensemble | 20 | | | |

### 4.1.1 Determination of the architecture of the MLPs

A comparison of the two target variables is performed by using the initial MLP architecture presented in Table 2, and Fig. 5 presents the results of this comparison. The ensemble of MLPs, with SWE as the target variable, shows slightly better values for the RMSE in Fig. 5d compared with Fig. 5a. Furthermore, in Fig. 5e the reliability part of the CRPS increases later when SWE is the target variable compared with when the target is variable density (Fig. 5b). This behaviour is also consistent with the ignorance score, which has its minimum at epoch 20 in Fig. 5f, whereas in Fig. 5c, the ignorance score for snow density increases from the beginning. This is consistent with the rank histograms in Fig. 6. The higher variability of SWE has a positive effect on the rank histogram, meaning that the spread of the ensemble remains sufficiently large up to epoch ten. From these results, we can derive that the lower variability of snow density results in too-little spread of the ensemble already after one epoch. This observation allows us to proceed with the target variable SWE. Nonetheless, even with SWE, the spread of the ensemble becomes too narrow, too quickly. This issue is addressed by incorporating input uncertainties, as we show in the following section.

We apply input uncertainty because we cannot train for a sufficient enough period to obtain the best error scores measured by RMSE, MAE, and CRPS. This inability is due to a loss of reliability and overfitting. This type of overfitting is related to

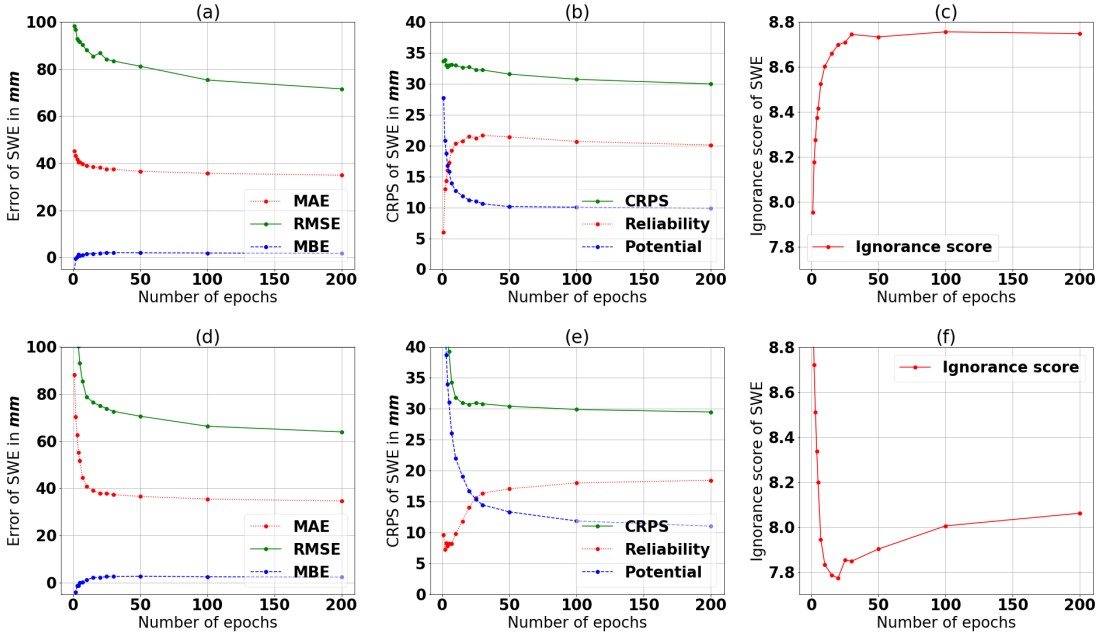

**Figure 5.** Comparison of (a)–(c) snow density and (d)–(e) SWE as target variables. (a) and (d) show the accuracy measures MAE, RMSE and MBE; (b) and (e) show the CRPS and its decomposition; (c) and (f) show the ignorance score; The scores are evaluated using the validation data set. We use the *Reference* setup presented in Table 2

the ensemble and does not describe the overfitting of one single MLP. To better understand, we examine the RMSE in Fig.
5, which is related to the MSE, the objective function in the optimization of the MLP. The RMSE continues to decrease beyond 15 epochs (optimal trade-off between accuracy and reliability). Therefore, the single MLP remains in an underfit state. Adding regularization would shift the MLP to a more underfit state by simplifying the network. Table 3 shows that incorporating input uncertainty decreases the ignorance score, whereas the other measures are unaffected. Therefore, input uncertainty widens the spread of the ensemble. Further improvements can likely be achieved by incorporating the input uncertainty of the
meteorological data through use of the ERA5 ensemble.

Table 3 presents all results for the tested characteristics listed in the row two to six in Table 2. The change of the activation function from tanh in the reference setup to ReLU and Leaky-ReLU does not produce any improvement. Using the ReLU function as the activation function in the network shows a similar but delayed behaviour in the performance scores. In terms of accuracy, we obtain similar results as with the tanh case via a longer learning. Nonetheless, the increase of the reliability part
is not delayed as much as the decrease in accuracy. This entails a worse trade-off between accuracy and reliability. When the activation function is set to the Leaky-ReLU activation function, we observe almost no change relative to the ReLU case. The analogous argumentation can be applied.

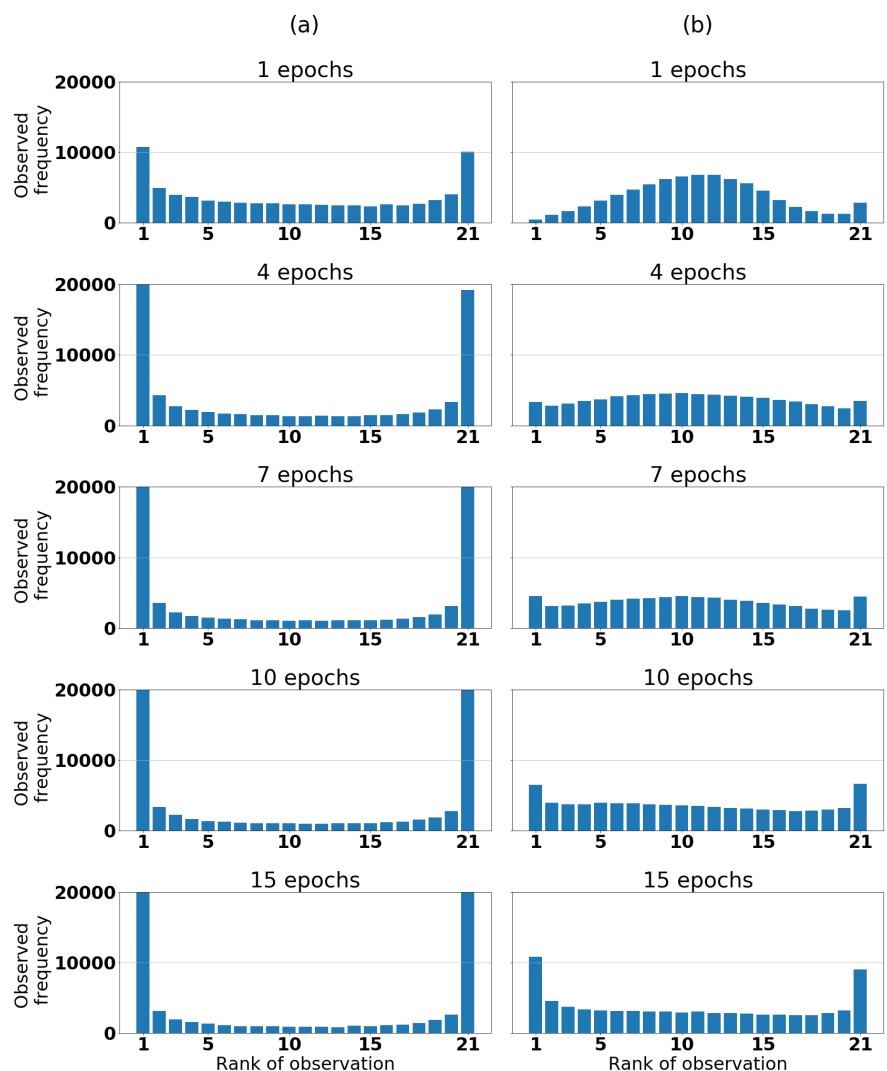

**Figure 6.** Rank histogram for different numbers of epochs (a) with snow density and (b) SWE as the target variables. We use the reference setup presented in Table 2

For the RMSProp algorithm, we set the global learning rate at 0.1, 0.01, 0.001, and 0.0001 during testing. The best result is produced for the trial having a learning rate of 0.0001. The learning rate of 0.001 produces an earlier decrease in accuracy scores and an earlier increase for the reliability part. The trials having a learning rate of 0.1 and 0.01 converge at higher values for RMSE and MAE. The higher learning rates can over-jump desirable minima and end up in worse ones. In summary, the overall learning rate must be small enough to achieve the best results. Even lower learning rates will entail a later behaviour. Therefore, the same trade-off between reliability and accuracy can be obtained if training time is increased. When we compare

**Table 3.** Comparison of all tested characteristics worth considering; Parameter initialisation [2] and [3] as in Table 2; The number of epochs is selected so that the trade-off between reliability and accuracy of the ensemble is increased. The underlined numbers in bold indicate an improvement throughout the testing. ***Combo*** combines para. init.[3], shuffled data, and input uncertainty, and is used as the final set up

| | Ref. | Input uncert. | ReLU | Leaky ReLU | RMSProp (rate 0.0001) | Shuffle data | Para. init.[1] | Para. init.[2] | Para. init.[3] | **Combo** |
|---|---|---|---|---|---|---|---|---|---|---|
| Epochs | 15 | 4 | 15 | 10 | 30 | 15 | 25 | 7 | 30 | **2** |
| MAE | 39.1 | 39.5 | 41.4 | 42.7 | 40.3 | 39.0 | 38.2 | 43.4 | 39.7 | **37.9** |
| RMSE | 76.5 | 78.5 | 82.3 | 87.3 | 78.1 | 75.7 | 72.1 | 80.2 | 79.4 | **76.4** |
| MBE | 2.1 | 1.9 | 1.7 | 1.7 | 1.0 | **0.4** | 2.8 | 1.6 | 1.9 | **0.3** |
| CRPS | 30.9 | 30.9 | 32.5 | 33.2 | 31.6 | 30.9 | 33.9 | 34.4 | 30.3 | **29.3** |
| Reliability | 11.8 | 11.5 | 11.7 | 10.4 | 10.6 | 11.9 | 25.2 | 13.6 | **7.5** | **8.9** |
| Ign. score | 7.79 | **7.49** | 7.83 | 7.86 | 7.81 | 7.76 | 8.35 | 7.94 | 7.77 | **7.4** |

RMSProp to AdaDelta, we note no improvement. Consequently, the AdaDelta method is preferred because there are no learning rates that need to be adjusted.

In regard to the parameter initialization, the equation [1] in Table 2 is a uniform distribution $U(-0.29, 0.29)$ in our application. The narrow interval results in a too-narrow ensemble, which entails a high value for the reliability part and ignorance score. The second trial, using the parameter initialization [2] (in our case $U(-0.68, 0.68)$), shows a very similar behaviour as the reference setup but with a lower number of epochs. This matches the interval of the random initialization being more narrow than that in the reference setup. It entails that the spread of the ensemble narrows more quickly and results in an early increase of the reliability part and an early minimum in the ignorance score at ten epochs. The third trial, using the parameter initialization [3], shows a delayed behaviour. However, this initialization has a better score for the reliability part of the CRPS but shows no improvement for the accuracy scores. We can conclude that to maintain a reliable ensemble, the interval of the uniform distribution must be sufficiently large. Nonetheless, an increase in the interval does not offer much improvement for the trade-off between reliability and accuracy.

Note that shuffling the data produces an almost identical result as that for the reference set-up. However, it eliminates bias, which is almost zero beyond 15 epochs.

### 4.1.2 Input variable selection

The results of the sensitivity scores for each explanatory variable are presented in Table 4. Recall that this sensitivity score is the average RMSE of the four reference percentiles and all ensemble members (see Sect. 2.4.7). Therefore, the value of this score is proportional to the importance of a variable for the conversion model. As shown in Table 4, snow depth is, unsurprisingly, the most important input variable. The model by Odry et al. (2020) uses six input variables and five out the six most important

**Table 4.** Ordered results of the sensitivity analysis from the least to most influential variable; The score is calculated following Sec. 2.4.7; The value of this score is proportional to the importance of a variable

| Variable | Score |
| --- | --- |
| Snow density ERA5 | 23.2 |
| Number of freeze-thaw cycles | 34.6 |
| Average temperature of the last six days | 63.9 |
| Number of layers in snow pack | 102.4 |
| Total precipitation in the last ten days | 102.5 |
| Average age of the snow cover | 116.9 |
| Accum. positive degrees since the beginning of winter | 124.9 |
| Accum. solid precipitation since the beginning of winter | 160.2 |
| Number of days since the beginning of winter | 184.6 |
| Total solid precipitation in the last ten days | 191.0 |
| Days without snow since the beginning of winter | 193.2 |
| Snow depth | 972.0 |

variables are coherent with the variable selection by Odry et al. (2020). However, the order of variables with scores lying close together can change, since the parameters are initialized randomly.

Figure 7 illustrates how reducing the number of input variables gradually affects the SWE estimation error and the corresponding CRPS and ignorance score. This reduction follows Table 4; the least influential input variable (snow density from ERA5) is removed first, and so on. Figure 7a and b show that the RMSE, MAE, and CRPS do not increase significantly when the number of input variables decreases to the six most influential variables. A larger increase (worsening) is only observed below this number. This pattern reflects the input variables being dependent on each other as all, except for snow density and snow depth, are derived from the same meteorological data. Therefore, adding more variables does not provide more information to the model. This result is consistent with the study of Odry et al. (2020). When observing the reliability part of the CRPS (Fig. 7b), the score begins to increase below ten input variables. However, the ignorance score in Fig. 7c increases even when only one variable is eliminated. Thus, more input variables help widen the spread of the ensemble. Nevertheless, the ignorance score tends to flatten as the number of input variables increases. Snow density from ERA5 shows the smallest importance and therefore, can be excluded for operational use. However, we want to preserve the spread of the ensemble and obtain a good portrait of the uncertainty of SWE estimation by using all 12 variables as inputs in the final setup.

### 4.1.3 Determining the number of epochs and the number of hidden neurons

First, we determine the number of hidden neurons and number of epochs for SMLP. Figure 8a–d show a general decrease in the MAE, CRPS, ignorance score, RMSE, and the reliability part of the CRPS when the number of neurons in the hidden layer

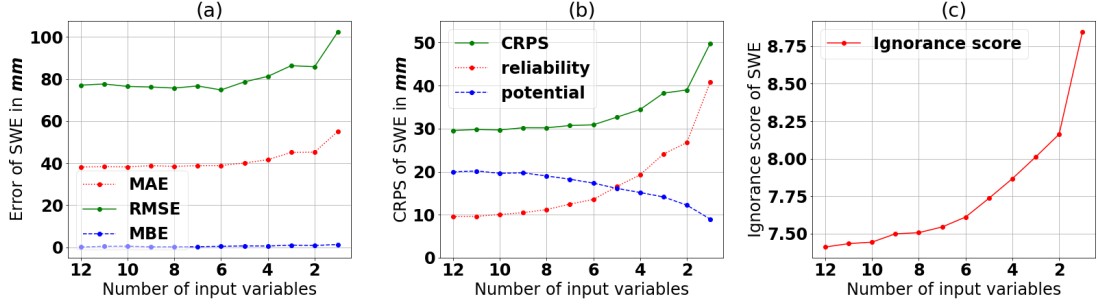

**Figure 7.** Stepwise reduction of input variables, ordered according to their influence as determined in Table 4. The scores are evaluated using the validation data set. The setup ***Combo*** from Table 3 is used;

is increased. A greater number of neurons in the hidden layer enables the model to learn more features of the training data set. This entails a better performance of the validation data set. However, there is a threshold beyond which these accuracy values start degrading. This threshold occurs at a smaller number of hidden neurons for a smaller number of epochs. For the MBE in Fig. 8f, we observe no specific trend. Note that the thresholds are earlier for the ignorance score than for the reliability part of the CRPS. Consequently, one can assume that the reliability will also increase for higher numbers of hidden neurons for three,

four, and five training epochs. The earlier increase in the ignorance score can be explained by observing the rank histograms of 1–3 training epochs in Fig. 9. A bell shape in the rank histogram indicates that the spread of the ensemble is too wide, resulting in low values overall for the empirical pdf. The ignorance score is computed using the logarithm of the probability density corresponding to the observation. Therefore, overdispersed ensembles are strongly penalized by this score. The formula of the reliability part of the CRPS in Eq. (9) demonstrates that the difference between the observed frequency, given the predicted

distribution, and the predicted probability is squared. Therefore, if the difference is small, this will dominate the reliability part and result in a small value even with a large ensemble spread. Thus, the reliability score penalizes the outliers in the trials having more than four or five epochs. Therefore, we set the number of epochs at five and the number of neurons in the hidden layer at 120. This provides the best trade-off between accuracy and reliability because 120 hidden neurons is the point where accuracy stagnates and where the ignorance score increases for five epochs. Except for the case of one training epoch,

the reliability part of the CRPS is poorest for five epochs, but it also leads to an acceptable rank histogram in Fig. 9e. The ensembles trained for two and three epochs show a bell shape, indicating underdispersion, which is an unacceptable condition.

      Second, we perform the same analysis for each snow class, to finalize the ensembles, one for each snow class for the MMLP model. Table 5 shows the optimal combination of the number of hidden neurons and number of epochs for each snow class. The behavior is similar to the previous case. However, as the amount of available data to train decreases, the number of required

training epochs increases. Also, snow classes with smaller datasets show smaller variability in the records, which can be easily represented by a simpler network with less hidden neurons, because of the lower complexity of the problem.

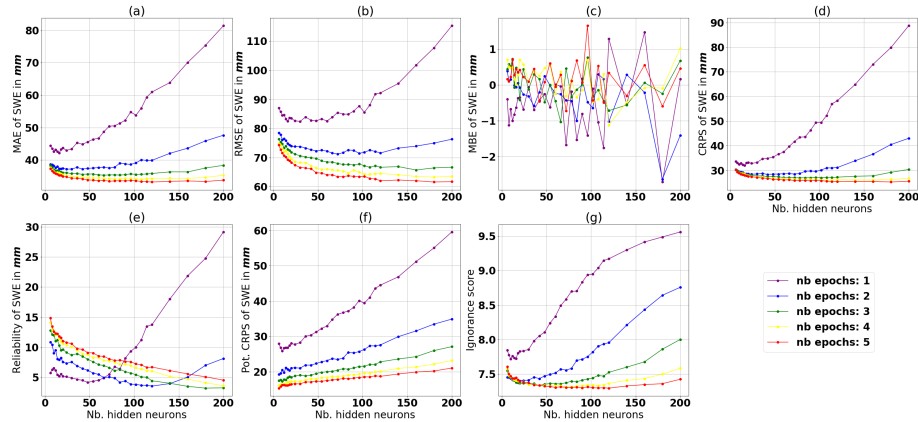

**Figure 8.** Results using the setup *Combo* from Table 3 for different numbers of training epochs and neurons in the hidden layer. It shows (a) the MAE, (b) the RMSE, (c) the MBE, (d) the CRPS, (e) the reliability part of the CRPS, (f) the potential CRPS and (g) the ignorance score. The scores are evaluated on the validation data set.

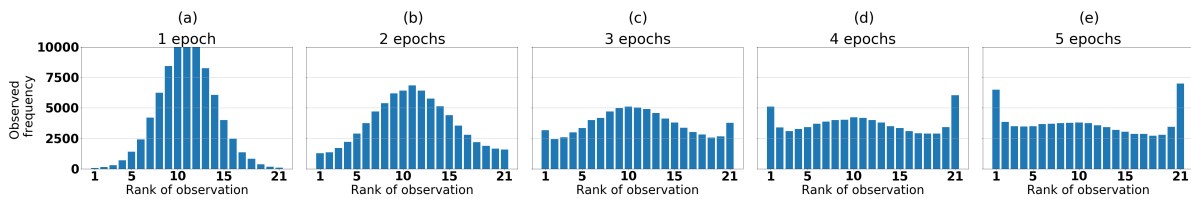

**Figure 9.** Rank histograms of the single MLP ensemble (SMLP) with 120 neurons in the hidden layer for different numbers of epochs, evaluated on the vaildation data set

### 4.1.4 Final setup of SMLP and MMLP

Table 6 shows the final setup of the two models SMLP and MMLP. The computational cost of training for the MMLP model is five times larger than that of to the SMLP model. The training was performed in 13 min and 62 min for the SMLP and MMLP model, respectively. The simulations on the testing data set take 3.5 sec and 18.1 sec for the SMLP and MMLP, respectively. The difference is mainly due to the loading process of six ensembles for the MMLP compared to one loading process for the SMLP, because the actual computation should be identical, except for assigning the records to the correct ensemble in the MMLP model. All computations were performed on a Dell XPS 13 9370 with an Intel(R) Core(TM) i7-85550U CPU processor of 8 GB RAM.

### 4.2 MLP model performance on a testing set

In this section, we use the testing data set and evaluate the performance of the SMLP and MMLP models. Table 7 reveals that the MMLP model has a better overall performance, as all performance metrics are smaller, except MBE. Fig. 10a and b show a

**Table 5.** Optimal combination of the number of hidden neurons and number of epochs for each snow class within the multiple ensembles model. Optimal combination is determined analogous to the single ensemble model in Fig. 8

|  | Ephemeral | Prairie | Tundra | Taiga | Mountain | Maritime |
|---|---|---|---|---|---|---|
| Nb. records (training data set) | 1124 | 3754 | 4451 | 7962 | 22414 | 38711 |
| Nb. epochs | 50 | 30 | 30 | 20 | 20 | 20 |
| Nb. neurons | 12 | 24 | 48 | 96 | 192 | 192 |

**Table 6.** Final setup of the MLPs for the single MLP ensemble model (SMLP) and the multiple MLP ensembles model (MMLP); The ensembles in the MMLP model differ only in terms of their number of hidden neurons and number of epochs. As a comparison, the setup of the ANN ensemble proposed by Odry et al. (2020) is presented as well.

| Characteristic | SMLP | MMLP | Odry et al. (2020) |
|---|---|---|---|
| number of ensembles | 1 | 6 (one for each snow class) | 1 |
| Target variable | SWE | SWE | snow density |
| Input uncertainty of snow depth | Yes | Yes | No |
| Activation function in the hidden layer | tanh | tanh | tanh |
| Optimization algorithm | AdaDelta | AdaDelta | Levenberg-Marquardt |
| Parameter initialization | $\sim U\left(-2,2\right)^3$ | $\sim U\left(-2,2\right)^3$ | $\sim U\left(-1,1\right)$ |
| Shuffling data before each epoch | Yes | Yes | No |
| Input variables | 12 | 12 | 6 |
| Number of hidden neurons | 120 | Table 5 | 6 |
| Number of epochs | 5 | Table 5 | 10 |
| Batch size | 100 | 100 | full data set |
| Number of members in the ensemble | 20 | 20 | 20 |

scatter plot where the median of the simulated ensemble is plotted against the the observations. Note that both models simulate negative medians for some data records. For the SMLP model, the minimum of the median is $-36\,\mathrm{mm}$ and for 0.6 % of the records in testing data set the model simulates negative SWE values. For the MMLP the minimum is $-42\,\mathrm{mm}$ and the ratio of negative simulation is 0.3 %. The reliability diagram, introduced in Sec. 2.4.5, shown in Fig. 10c and d, reveals a closer fit to the identity line for the MMLP model and therefore, a more reliable estimate of SWE. The same conclusion can be drawn from the rank histograms presented in Fig. 10e and f. As mentioned in Sec. 2.4.4, a flatter rank histogram indicates a more reliable estimate.

**Table 7.** Performance evaluation of the model with a single MLP ensemble covering Canada (SMLP) and the multiple MLP ensembles (MMLP), evaluated using the testing data set

|  | SMLP | MMLP |
|---|---|---|
| MAE | 32.8 | 29.3 |
| RMSE | 61.0 | 51.5 |
| MBE | 0.4 | 0.6 |
| CRPS | 25.1 | 21.9 |
| Reliability | 6.4 | 3.7 |
| Pot. CRPS | 18.6 | 18.2 |
| Ign. score | 7.28 | 7.22 |

Figure 11 shows the distribution of the residuals of the simulated ensemble median and the observations. The distribution of the residuals for the multiple MLP model is narrower, indicating less error overall. Both are approximately symmetrical around zero, which strengthens the result of the MBE being almost zero (Table 7). For the SMLP model, 50 % of the errors are between $[-17.1, 18.4]$ mm, and 90 % are between $[-77.6, 73.3]$ mm. For the MMLP model, 50 % of the errors are between $[-15.8, 18.0]$ mm, and 90 % are between $[-68.8.1, 68.6]$ mm.

Next, we apply skill scores to ensure a valid comparison between SWE estimates of differing magnitudes. The climatology of SWE from the CHSS data set is used as the reference simulation (see Sect. 2.4.6).

The separation into different snow classes reveals that, as expected, the multiple MLP model eliminates bias in all snow classes, as shown in Fig. 12a. Furthermore, for all snow classes, the two MLP models perform better than climatology, as indicated by positive MAE, RMSE, and CRPS skill scores in Fig. 12a and b.

Furthermore, Figure 12a shows that both models perform best within the *ephemeral snow* class in terms of accuracy, although the results need to be taken with some reservations; the low amount of data and the high variability of SWE in this snow class induce a rather poor reference simulation (climatology). Also, the reliability part of the CRPS in Fig. 12b shows a large improvement of the multiple MLP model for the *ephemeral snow* class. In contrast, the ignorance score in Fig. 12c shows a slightly poorer skill score for this same snow class. A deeper analysis of the *ephemeral snow* class in Fig. 13 shows that the MMLP model leads to an almost reliable rank histogram, with only a slight tendency toward underdispersion, whereas the single MLP model shows a overdispersed rank histogram. This pattern explains the large improvement in the reliability part of the CRPS. The average spread of the ensemble for the multiple MLP model in this class is twice that of the single MLP model. The larger spread entails lower values in the empirical pdf and therefore lower ignorance scores.

The *tundra snow* and *taiga snow* classes have the lowest skill scores in terms of both accuracy and reliability. In particular, the reliability part of the CRPS in Fig. 12b results in a large decrease of the skill score. As the variability of SWE is low in

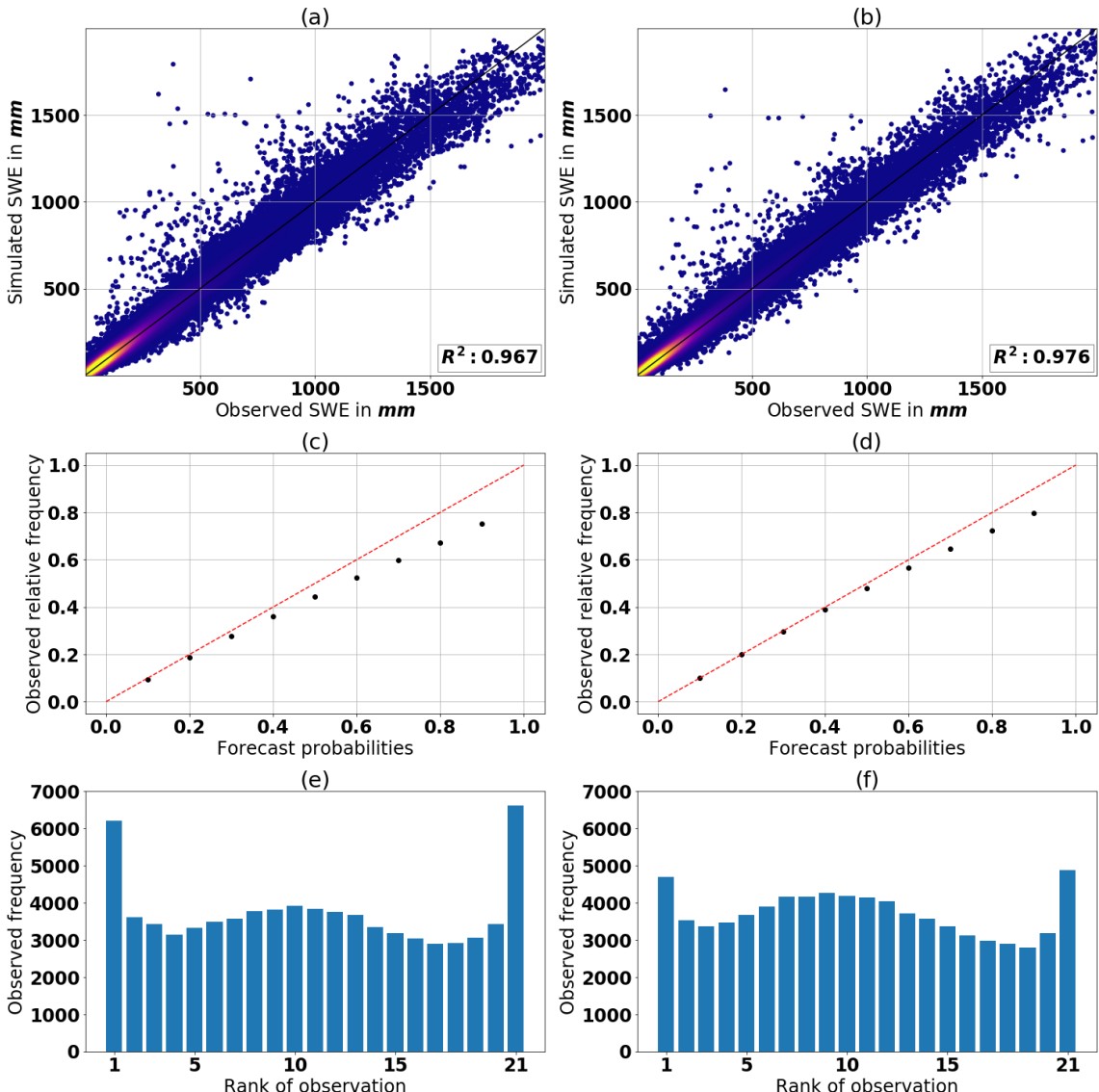

**Figure 10.** The scatter plot in (a) and (b) shows the median of the simulated ensemble against the observation for the single MLP ensemble (SMLP) and the multiple MLP ensembles (MMLP), respectively. Note that the axes are cut of at 2000 mm. Zoomed out scatter plots are shown in Fig. 15. The reliability diagrams are presented in (c) for the SMLP and in (d) the MMLP. Further, the rank histograms are presented in (e) for the SMLP and in (f) the MMLP. All shown results are for the testing data set

these areas, as presented in Fig. 2, the reference simulation based on climatology produces a reliable ensemble and causes a poor reliability skill score for the actual simulation.

The *maritime snow* class shows a slightly better accuracy than the *mountain snow* class. This difference may relate to the complexity of snow accumulation patterns in mountainous regions owing to the high spatial and temporal variability of all

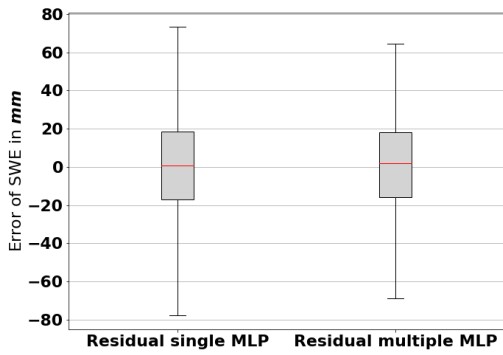

**Figure 11.** Distribution of the residuals of the simulated ensemble median and the observations. The boundary of the box shows the 1st and 3rd quartiles; the caps at the end of the whiskers show the 5th and 95th percentiles

physical processes and variables in these areas. Also the temperature correction, as explained in Sect. 3.2, induces larger errors where height is highly variable. The improved reliability skill score for the multiple MLP model is explained by fewer outliers for the *mountain* and *maritime snow* classes. The slight decrease in the ignorance skill score for the *maritime snow* class can be explained by different average spreads of the simulated and reference ensembles.

In the next step, the testing data set is divided into elevation classes from 0 m to 2100 m, with a step size of 300 m and
580 one class for sites above 2100 m. The results are not shown, as the main conclusions are identical to those obtained from the separation into snow classes. Both types of MLP ensembles (single and multiple) outperform climatology.

Furthermore, an analysis over the course of the year shows an improved accuracy for the MLP models compared with climatology for all months, except for July, August, and September. During these three months, there is generally very little snow across the country and, consequently, very little data. Additionally, the *beginning of the winter*, subjectively taken as
01 September, causes a reset to zero for several input variables; the variables of temperature, snow depth, and SWE data remain, however, within their usual ranges. This greatly complicates the proper training of the MLPs. Overall, except these three problematic months, the accuracy and reliability remain relatively constant throughout the year with a slightly improved performance in spring and early summer compared with climatology.

### 4.3 Comparison of MLP models with regression models

The regression models are trained and validated using the same perturbed snow depth data sets that are used to optimize the MLP models. The testing data set is then used for comparison purposes in this section.

There is a possibility of producing an ensemble simulation by performing the deterministic regression models on perturbed snow depth and obtaining multiple members. However, given that the spread in the simulated ensemble of the MLP models is explained mainly by the various parameter initializations, this approach entails an ensemble that is too narrow when regression
models are used. This leads to a comparison of models that has little meaning: the optimization of the Sturm model could be

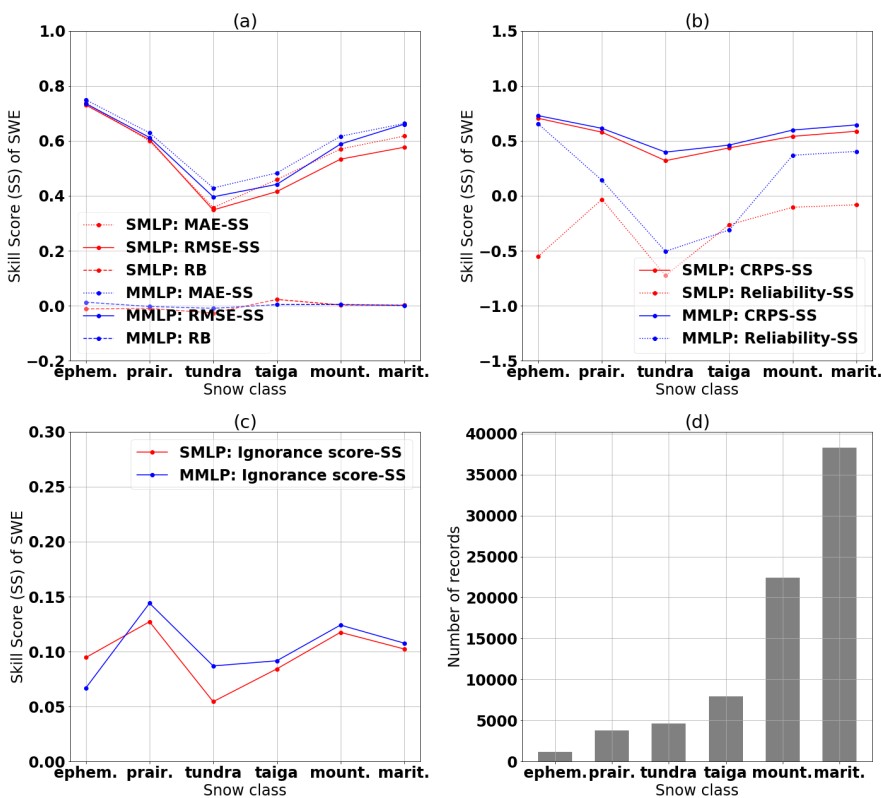

**Figure 12.** Comparison of the performance of the single MLP (SMLP) and multiple MLP (MMLP) models using (a) the skill score (SS) of the MAE, RMSE and the relative bias (RB), (b) the skill score of the CRPS and reliability and (c) the skill score of the ignorance score evaluated for the different snow classes. All results are for the testing data set. (d) The number of records in each snow class within the testing data set

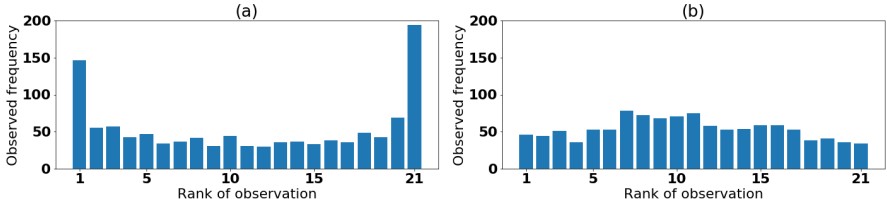

**Figure 13.** Comparison of the rank histogram of (a) the single MLP (SMLP) and (b) multiple MLP (MMLP) for the *ephemeral snow* class using the testing data set

initialized with different parameter sets; however, the Jonas model has a perfect set of parameters. Therefore, we must compare the models using exclusively deterministic evaluation techniques.

**Table 8.** Comparison of the overall performance evaluation using deterministic performance evaluation metrics of the two MLP models and the regression models, evaluated using the testing data set

|      | Benchmark | Jonas | Sturm | SMLP | MMLP |
|------|-----------|-------|-------|------|------|
| MAE  | 74.2      | 44.5  | 47.7  | 32.8 | 29.3 |
| RMSE | 145.4     | 114.7 | 117.0 | 61.0 | 51.5 |
| MBE  | -35.3     | 2.2   | -0.5  | 0.4  | 0.6  |

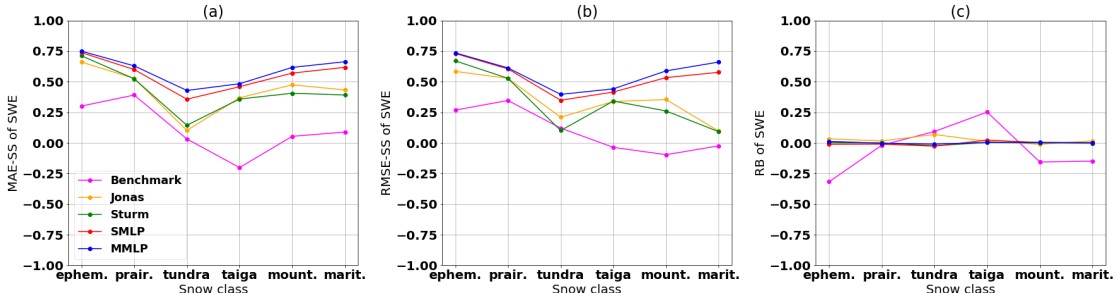

**Figure 14.** Comparison of the performance of the two MLP and regression models by using (a) the skill score of the MAE, (b) the skill score of the RMSE, and (c) the relative bias (RB) of the SWE simulation, as evaluated for different snow classes. Results are shown for the testing data set.

In addition to the regression models, we also considered a simple benchmark model, which takes the average observed snow density in the training and validation data sets as a constant snow density and then calculates SWE for the testing data set.

600  The results are presented in Table 8. Both models using MLP ensembles outperform the simple benchmark as well as the Sturm and Jonas models.

  Figure 14 compares the MLP models with the regressions and the simple benchmark for each snow class. The two MLP models outperform both the regressions and the benchmark model for all snow classes. The Sturm model eliminates the bias, as depicted by the relative bias (RB) being equal to zero in Fig. 14c. This removal of bias occurs because the model uses an

605 individual regression model for each snow class. In Fig. 14a and b, the MAE and RMSE skill scores of the Sturm and Jonas models follow mainly the skill scores of both MLP models. However, the Sturm and Jonas models show a decrease in the RMSE-SS for the *mountain snow* and *maritime snow* classes, which indicates a greater number of outliers. One can derive that a single regression model is not sufficient for the high variability in the mountainous region or the spatially large maritime region, which includes two coasts. The better performance of the Jonas model for the *mountain snow* class reflects this model

610 being built and tested for a mountainous region in the Swiss Alps. The benchmark model performs well in the Prairies; this result indicates a stable relationship between snow density and snow depth in this snow class. The poor performance of the *taiga snow* class for the benchmark model is caused by the high positive RB of the benchmark model, as presented in Fig. 14c.

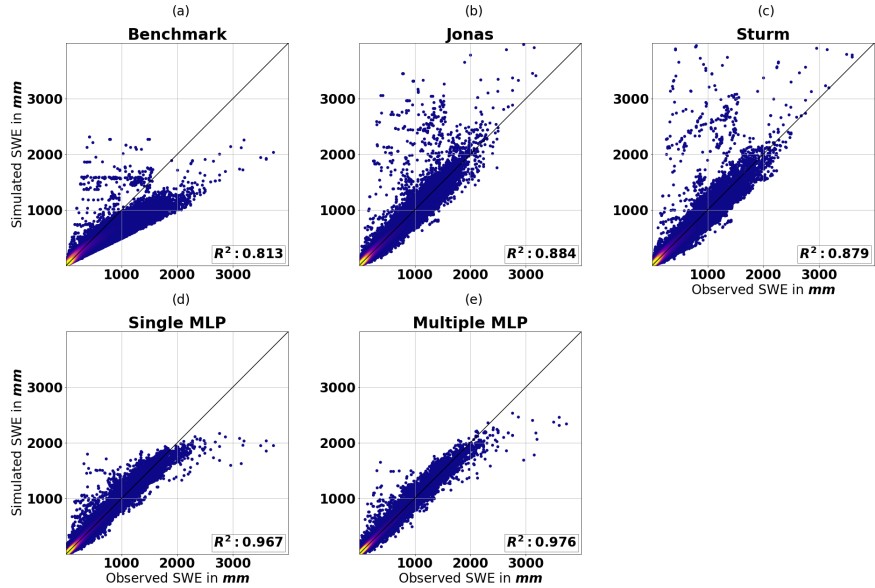

**Figure 15.** Scatter plot of median of the SWE ensembles and observed SWE for each model, evaluated on the testing data set; note that the axes are cut off at 4000 mm; thus, 0.03 % and 0.01 % of the data points are not shown for Jonas and Sturm model, respectively;

Figure 15 shows the simulated SWE against the observed SWE. The benchmark model can explain 81.3% of the variability in SWE. Figure 15a shows that the model often underestimates SWE, which also explains the negative bias observed in Table 8. The Jonas and Sturm models explain approximately 7 % more SWE variability. Figure 15b and c reveal that outliers with high values occur for lower- and medium-range SWE observations. Note that the axes are cut off at 4000 mm. Thus, 0.03 % and 0.01 % of the data points are not shown for Jonas and Sturm model, respectively. The 17 SWE observations above 2500 mm in the testing data set are consistently overestimated but follow the identity line. Compared with the two MLP models, the Sturm and Jonas models perform better at these higher values. The training data set contains 18 SWE measurements above 2500 mm. This small portion makes it problematic for any model to properly train on. However, it seems that the linearity of the Jonas model and the log-linearity of the Sturm model can extrapolate high SWE values better than the MLP models. Overall, compared with the benchmark model, the single and multiple MLP models explain approximately 15 % and 16 % more SWE variability, respectively. Figure 15d and e show that up to 2500 mm, the estimates of both MLP models follow mainly the identity line with a some outliers at approximately 500 mm. Above 2500 mm, the model seems to have an upper boundary that results in an underestimate. This upper boundary is less restrictive for the multiple MLP model, which performs better for the high SWE records. The resulting $R^2$ values for the benchmark and Sturm model are similar to those in the study of Odry et al. (2020), which considered only the province of Quebec. The MLP model in the latter study achieved an $R^2$ value of 0.919. Therefore, we improve on this by 5 % and 6 % for the single and multiple MLP models, respectively.

# 5 Conclusions

This study tackles some important knowledge gaps regarding the conversion of snow depth to SWE, using ANN-based models. The main focus is on the architecture of the network, and two hypotheses are tested. The first hypothesis holds that using SWE rather than density as the target variable for the ANN will produce more accurate estimates of SWE. The second hypothesis states that in-depth testing of several ANN structural characteristics (e.g. optimization algorithm, activation function, parameter initialization, increasing the number of parameters) can improve the estimates of SWE. We thus investigate whether the ANN model must be trained specifically for different regions, as determined by snow climate classes (Sturm et al., 2009), or whether the model could be trained only once, for Canada as a whole. The uncertainty of snow depth measurements is included. Furthermore, we use existing regression models, developed for the same purpose, as benchmarks to obtain a better perspective on how our model performs. We were able to find a structural configuration of the ANNs that leads to noticeable improvements compared to the initial basic configuration proposed by Odry et al. (2020). A final comparison is given in Table 6. Therefore, hypothesis 2 is also verified, at least for the available data in Canada.

Our snow-depth-to-SWE model uses the inputs of snow depth, estimated snow density, and other explanatory variables derived from meteorological data. The available snow data includes snow depth, SWE, and snow density measurements from across Canada, collected over almost 40 years.

We then use an ensemble of multiple MLPs to address the issue of the random parameter initialization during optimization. The approach also provides a probabilistic estimate to gain greater insight into model performance. A trade-off between reliability and accuracy is used as a means of evaluation, which gives a more comprehensive analysis of SWE-estimation models.

Many previous models (e.g. Odry et al., 2020; Jonas et al., 2009; Sturm et al., 2010; Painter et al., 2016; Broxton et al., 2019) determine SWE by estimating snow density and calculating SWE on the basis of snow density and snow depth. This study investigates a direct estimate of SWE. This approach shows a slight increase in accuracy and a large gain in reliability compared to indirect estimates, as presented in Fig. 5 and 6. Consequently, the study uses SWE as the target variable rather than snow density.

In our investigation of model structures, we built two models. One model uses a single MLP ensemble for all of Canada. The second model trains one MLP ensemble for each snow class, as defined by Sturm et al. (2009). Model evaluation of the independent testing data set indicates that the multiple ensemble model outperforms the single ensemble model. Both models show weak performances for high values of SWE. Furthermore, both MLP models outperform existing regression models and a benchmark model based on climatology, and they improve on the basic MLP ensemble model proposed earlier by Odry et al. (2020). Also, the current study uses a broad snow data set with records of all snow classes (except the *ice snow* class, for which we have too little records for a proper analysis) and thus considers diverse snow characteristics across a large domain. Therefore, the model structure is expected to be applicable to other areas in the world. However, new training is advisable.

A sensitivity analysis reveals that a greater number of input variables increases the reliability of the ensemble. Therefore, adding more variables could further heighten the model's reliability. After proposing SWE as the new target variable in this

study, short term and long term variables regarding precipitation with respect to SWE need to be analyzed. In Table 1, different correlation were found for SWE and snow density, when looking at short term variables for 1 to 10 days. Odry et al. (2020) showed in their Table 4 that snow density is negatively correlated with recent snow fall, because it is lighter than the underlying snow. This negative correlation peaks at 3 days for the data set used in this study (Table 1). The correlation between SWE and accumulated solid precipitation increases for variables of larger time range due to diverse densification factors, resulting in a more stable relationship between the two variables. Therefore, a deeper analysis of the information content carried by recent accumulated solid precipitation with respect to SWE is favorable. Furthermore, topological variables such as the slope and aspect of measurement site can be used. This might even improve the estimation accuracy, because these variables carry additional information compared to input variables derived from meteorological data. To test the the ability of ANNs in itself, the model can be trained on more precise meteorological data on a limited number of sites close together to ensure a consistent data set.

Regarding the limitations of this study, both models show poor performance for high SWE values, mainly because the amount of available training data is low for those extreme values. Furthermore, the model is not predictive and especially cannot account for the effect of climate change. It is noted that the models requires more data compared to the regression models proposed by Sturm et al. (2010) and Jonas et al. (2009), because multiple indicators are calculated from temperature and precipitation time series. The amount of data needed to train the model properly cannot be prescribed universally, as it depends on the variability of the data set. For instance, if an area shows many different snow classes, more data is needed to obtain satisfactory results. This also changes the number of epochs and number of neurons needed. Since the testing of different characteristics in Sec. 4.1.1 showed only little improvement, we expect that the determination of the importance of input variables and a cross analysis of number of hidden neurons in the network with number of training epochs using a validation data set is sufficient to obtain satisfactory results.

As mentioned in Sec. 4.1.2, snow density would not be used in operational use. Furthermore, the recently available Regional Deterministic Reforecast System (RDRS) would be used for the meteorological data when training the model. RDRS has similar dynamics and physics as the operational Global Environmental Multiscale Model (GEM). This model could then be used to simulate SWE from in-situ snow depth measurements by sonic sensors provided by Meteorological Service of Canada and ECCC (2020). As meteorological data, one could use an operational "nowcast" from an atmospheric model that include a land data assimilation system such as the Canadian Land Data Assimilation System (CaLDAS; Carrera et al. (01 Jun. 2015)). CalDAS is forced by real-time precipitation analyses from the Canadian Precipitation Analysis (CaPA; Fortin et al. (2015)), which combines simulated background precipitation fields with observed data (in situ and radars). Furthermore, the proposed method can be applied onto assimilated snow depth data in CaLDAS. Currently in CaLDAS, only snow depth data is assimilated and subsequently converted to SWE using the simulated density to initialize the land surface scheme. The proposed method would allow for two important upgrades: First, it would allow to assimilate snow depth data (converted to SWE) as well as SWE data, thus increasing the quantity of assimilated observations, and second, it would avoid using the simulated density, which is very hard to simulate accurately.

Finally, this study shows an optimal performance of networks having large numbers of neurons in the hidden layer at amounts far above the commonly used rules of thumb. This provides a motivation to look into network structures having multiple layers. Montúfar (2014) also showed that the number of neurons needed to approximate a function in a deep network with multiple layers increases exponentially in a network with one hidden layer. Therefore, deep networks are possibly more efficient. Goodfellow et al. (2016) provides multiple examples that foster this claim empirically. As a result, there could remain numerous means of improving SWE estimates from snow depth using machine learning techniques, and the methods proposed here could be refined. It could also be useful to investigate the application of other types of machine-learning algorithms, including random forests.

*Code and data availability.* The code including some testing data is available on *GitHub*. The whole data set, excluding data for which we did not have permission to share (this is 2% of the data set) is available through the *Harvard Dataverse*.

*Author contributions.* Konstantin Ntokas (K.N.) performed all the computations, suggested most of the specific tests to be undertaken, and wrote the initial version of the manuscript. Jean Odry suggested edits to the manuscript and provided the initial version of the codes to K.N., which K.N. then translated to Python and modified. The original idea for this work was from Marie-Amélie Boucher, who also guided the work throughout and edited the manuscript significantly. Camille Garnaud was involved in the guidance for this project as a representative of ECCC and provided a final proofread of the manuscript before its submission.

*Competing interests.* The authors declare no competing interests.

*Acknowledgements.* The authors acknowledge the financial support of Environment and Climate Change Canada and are also thankful to the Réseau Météorologique Coopératif du Québec for providing the data required for this study. Furthermore, the authors would like to thank Vincent Fortin and Vincent Vionnet for their contribution and input throughout the project.

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
