# Peer review of "Investigating ANN architectures and training to estimate SWE from snow depth"

_Hydrology and Earth System Sciences, 2020_

## Referee Comment (RC1) · Anonymous Referee #1 · 23 Dec 2020

**General Comments**

This manuscript describes a novel method for estimating snow water equivalent from snow depth and a variety of other variables, by using an ensemble of artificial neural networks (ANN). This type of machine learning model is becoming increasingly popular as computational power increases. Likewise, the estimation of snow water equivalent from remote sensing or other in-situ variables is in constant development due to the scarce availability of in-situ SWE measurements. The manuscript falls within two topics of current great scientific interest and within the scope of the journal.

I first have to congratulate the authors for the huge amount of work that they have

done in the analyses and the text. This is a follow-up study to Odry et al. (2020), who introduced the use of ANNs to estimate snow density from snow depth over Quebec, outperforming other regression models such as Jonas et al 2009 and Sturm et al 2010. Here, the authors develop and improve the initial model. As elements of novelty, the authors: (1) perform an in-depth analysis of model architecture, testing several options of model characteristics; (2) demonstrate that training a model for different snow classes improves model performance; and (3) use SWE as target variable instead of snow density, which also improves performance. In general, the text is well structured and well written. The introduction and literature review are lengthy but are useful for a non-specialised reader to understand the theory behind multilayer perceptrons (MLPs) and the model ensemble evaluation metrics. The experimental set up is very detailed, although it lacks some clarity in its structure. The results are extensive and support the conclusions reached by the authors. However, the authors should be more convincing about the scientific progress that their analyses provide, as compared to Odry et al. 2020. Moreover, the results are generally very descriptive but some more discussion (or a discussion section) lacks. The results and conclusions are focused on the model performance improvement, but there is no discussion about the limitations of these type of models (e.g. the amount of data or computation time they require). There are also quite a few specific issues to be addressed to improve the quality, clarity, and reproducibility of the study. Overall, the work presented in this study is impressive, but the authors should address the issues that I outline here before I can recommend it for publication.

**Specific comments:**

1. Lines 74-79: Here you should state more clearly what the novelty of this paper is compared to Odry et al. 2020. Before I finished reading the entire paper, I was not convinced that there would be enough novelty in this manuscript. Perhaps a good solution would be to add a few lines before this paragraph, summarising the key knowledge gaps that you are filling (testing model structures, target variables, including climate

classifications, …), and why it is important to tackle these knowledge gaps. What is the aim of the paper besides just "improving a model"?

2. As estimated from line 309 and Figure 3(e), snow depth-SWE measurement sites have on average 100 records. This means that the model is mostly developed over independent and sparse measurements, rather than with time series. The influence of this is not discussed. Would MLPs benefit from training with long term time series? What would be the influence of including back-propagation loops in that case? Please discuss it. Testing it for a long-term dataset such as the SNOTEL dataset over the United Stated would be worth, but I acknowledge this would require an entire follow-up analysis.

3. The explanatory variables in section 3.2 are identical to those in Odry et al. 2020, with the addition of snow density from ERA5, which turns out to be the least explanatory in Table 5. Why didn't you include other variables suggested in Odry et al. 2020 such as wind and solar radiation?

4. For variables of accumulated precipitation in the last n days, n is only tested for 1 to 10 days. In Table 1, three out of four variables show largest correlation for 10 days, which makes me suspect that the range tested is not big enough. Please test for a larger range. In case this becomes increasingly large, it could be that accumulated precipitation in the last n days and accumulated precipitation since the beginning of the winter are the "same explanatory variable".

5. The structure of Section 3 is somewhat confusing. I don't clearly understand what is a "tested characteristic" and what is not. For instance, input uncertainty and input variable selection are two "characteristics" that are tested according to Table 2 and 3, but they have their own subsection (3.3 and 3.5 respectively), while the other tested characteristics (e.g. optimization algorithm) are described in section 3.4. I suggest blending section 3.5 and 3.2 together, and to include 3.3 and 3.6 in 3.4. Combining Table 2 and Table 3 together would also help understand what is being tested and

what is not. Please either restructure the section, or clearly justify the current one. The results in section 4.1 should then be restructured according to the new structure of section 3.

6. Table 4: the combination (combo) of the best choice for each tested characteristic provides even better performances for most evaluation metrics. Why are all the characteristics tested one by one? Please test the combined effect of characteristics or provide a clear explanation about why this is not possible (computationally too expensive?) or not necessary.

7. Line 472-473: What part of Section 4.1 are you referring to? It is not even finished, 4.1.6 is still coming after this. I suggest to provide an extra table (or a paragraph) with the final MLP architecture set-up, to avoid having to jump section by section to gather all the final decisions.

8. There is barely any discussion about computational cost of the final model architecture set-up, and about the number of epochs and hidden neurons. If applicable, I would like to see what the computational trade-off of the final model choices is (e.g. choosing for XXX is about 103 times slower than YYY, though I acknowledge this is CPU dependent).

9. Line 482-483: and the number of neurons decreases too. Please explain why.

10. Lines 489-491: Why and what does it mean? This is the only reference to Figure 10 in the text. Please provide more information on the results shown in the figure. A discussion of the results is also necessary here, since there is no Discussion section.

11. Table 7: How are the values obtained for MMLP? Is it the median of the ensembles for each snow class? Please specify.

12. Figure 10: top row: How is the "median simulation" obtained for a specific bin? How is the histogram of medians built? This is highly unclear, please specify. Also, why is there a small bin on the negative side? Does the model simulate negative SWE

values? Middle row: Is this a reliability diagram? It can be guessed from the text but it is not specified anywhere else. Please provide a letter for all subpanels (a-f).

13. Lines 495-496: Looking at figure 11 the values seem to be rather +/-18 and +/-16? Please provide the accurate values.

14. Line 552: In what figure/panel do you see the better accuracy for ephemeral snow? Please add this information.

15. Line 507: Where is the deeper analysis of the ephemeral snow class shown? The text refers to a rank histogram for the snow class, but I don't see it. Same in line 520 referring to a rank histogram for mountain and maritime snow, where is it? Please show them.

16. Figure 12: I believe the Y axis in panel (a) should be Skill Score (SS). Just as you did in Figure 13, please write "Skill Score" either on the axis label or in the caption, it is useful for the reader to be reminded what "SS" means. Same with RB, I had to read the text to find what it means (Relative Bias). Why the SMLP is evaluated on RB and MMLP on MBE-SS?

17. Figure 14: Given the large amount of data, consider adding colour to the scatter plots based on the density of points.

18. Line 562: How many are "numerous"? It seems as if they are <10? In that case, the MLP probably can't be trained for those high values of SWE, while the regression model is continuous also for high values. Please discuss the lack of training data for these high values.

19. TITLE: The title of the manuscript is still too similar to Odry et al. 2020. Think of a title that would directly show the improvement/novelty with respect to Odry et al. 2020. (e.g. use "testing ANN architectures", "snow class", "climatological variables" or "multiple ensembles")

20. CAPTIONS: Throughout the manuscript, Figure and Table captions are just one

line long. Please extend them. Make sure everything (acronyms, lines, points, etc.) that is shown in the Figure or Table is described on the legend, labels, or in the caption.

21. DISCUSSION: Following many of my comments, provide discussion of results within the results section, or provide a separate discussion section before conclusions. I especially miss a discussion on limitations of the model, applicability and transferability. How much data does the model need in order to be properly trained? You could for instance re-do the analyses but using only a random 10

**Technical corrections:**

Line 75: Please replace "verify" for "test", otherwise it seems that the hypotheses have been customised based on your results.

Line 192: I'm guessing this is a typing error, otherwise why is DOYobs = 0 on 1st January? Shouldn't it be 123 provided 1st September is DOY = zero?

Line 457: Figure 8a-e

Figure 8: panel g should be f, and h should be g. Also, I suggest making a 4x2 (or 2x4) panel figure, with the only empty panel filled with the legend. It would reduce white space.

Line 526: "Second, [. . .]" but where is first? Rephrase accordingly.

Line 588: Remove "the remainder of".
* * *

---

## Referee Comment (RC2) · Anonymous Referee #2 · 6 Jan 2021

This manuscript describes the application of machine learning techniques, specifically an ensemble of multilayer perceptrons, to estimate the hydrological variable Snow Water Equivalent (SWE). As described by the authors, SWE is a crucial variable which is difficult to directly measure at scale. The paper subject advances the application of ML techniques for SWE estimation by application of ensemble methods, discerning model applicability over climatic regions. and demonstrating advantages over empirical methods.

This research builds on the Ordy et al (2020) study by extending the geographical scope, using direct estimation of SWE and introducing snow classes in MLP training. There is sufficient novelty in this paper for publication, however it should first be strengthened in clear justification for decisions, interpretation of results and conclusion.

The manuscript leaves out some essential elements from Ordy et al, including descriptions of evaluation metrics and why they are selected. The addition of explanatory variables such as Snow Density from ERA5 lack explanation and background. Following strong results reporting sections, the discussion finding and conclusions of the manuscript require additional reflection on the limitations of the study and the context.

Overall, a strong effort worthy of publication on the basis of some revision and structural improvement. Some coherence is missing in the experimental design, in the inclusion of variables, the applicability of the study and the conclusions drawn from it. These require revision, hope the comments that follow can be of help.

Pg 1, ln 15: "Using a greater number of MLP parameters could lead to further improvements" It is somewhat self-evident that increased parameterization of an MLP model could potentially produce better results, can this statement be focused to the study specific outcomes?

pg 2, 50: This description of the application of physics-based models for SWE estimation is a bit too simplistic here, given ERA5 snow density as used later as an explanatory variable. The iSnobal mentioned is a coupled energy and mass-balance model that requires a great deal of meteorological data derive accumulating snow density and in turn modelled SWE. Please provide some further description on the advanced requirements these approaches and limitations, beyond only computational cost.

pg 2, ln 54: Consider including Snauffer et al, 2018. https://doi.org/10.5194/tc-12-891-2018.

pg 3, ln 76: The second hypotheses seems too broad. "in-depth testing". Please be more specific as to the methodology to be tested.

pg 3, ln 78: "The entire area of Canada." Is this an overreach given the the limited

density of measurements across much of Canada? "Applicability in a broader context". Be more specific in the what this broader context is.

pg 3, ln 78: This last sentence seems out of place to close the paragraph. Moving one sentence earlier would improve the paragraph.

pg 3, ln 94: Is MSE the definitive objective function for regression problems? Better likely to phrase as "commonly used"

pg 5, ln 131: "The algorithms RMSProp and AdaDelta produce good results". Please elaborate in this statement, or tie in better with the following two sentences.

pg 5, ln 155: Avoid starting sentences with "Because". This is a general comment also through the manuscript. Would recommend re-writing this initial sentence, breaking into sections.

pg 6, ln 171: Can references be provided for some of these conclusions? The linkage of snow depth only to precipitation requires some basis. There are a lot of varied physical processes for snow accumulation between tundra and taiga eco-zones.

pg 8, ln 222: I don't follow the second sentence, though can be my ignorance. Consider a clearer explanation if including.

pg 12, Figure 3e. It is possible to rescale the number of records for each site? It is not very descriptive. Is the maximum records up above 3000?

pg 13, Ordy et al had recommended the inclusion of additional explanatory variables from meteorology, such as wind or solar radiation. This study has included ERA5 daily averaged snow density data. This output from a physically based model is included without description of its generation, assessment of the quality or the relevance or applicability of this data source. Although the variable is kept as least important for the conversion model, and minimal impact on the ignorance score, it is kept in complete assessment. The manuscript should include some rationale for the inclusion of this model output, and why it was chosen. Are the assumptions in producing the snow

density relevant? How does this data perform compared to available measurements?

Pg 15, Table 2, To clarify, on pg 4, ln 97 it is mentioned that modifying the order of the input data is recommended. Is that done in this study (Shuffling data before each epoch)?

Pg 17, Figure 5. MBE should be introduced before used. The use of error metrics is not entirely clear (MBE, MAE, RMSE) compared with clearly rational and description for other scores. Clearer rationale and explanation would help. Is there a reason the RMSE is shown compared to the objective function MSE? RMSE can be a more comparable error metric for SWE, but this is not explained.

Pg 20, line 449: This appears a notable and relevant finding (what explanatory variables are ultimately useful) that can be better articulated in study findings.

Pg 27, Figure 13: Consistency would be useful for interpretation between RB and MBE. Can see in the following paragraph why RB is substituted for MBE, but would like to see this graph included.

Pg 27, line 555: The conclusions drawn in this section appear to have a relatively weak causal or testable links. Ranging from regression model structure, to physical processes to reference model performance, several comments seem quite speculative. For example, the tundra region has poor performance, but would be subject to may be similar topographic controls of the prairies. It would seem better to reflect on what information can truly be derived from these results, or at least address that there are many contributing factors that are not represented by this method.

Pg 28: line 570: This opening sentence for the Conclusions section should be more descriptive and engaging in the content of the study.

Pg 29, ln 591: What is the additional geophysical information beyond snow class from Sturm et al.? If this refers the discretization by elevation class, this should be elaborated on in the rest of the document to include in conclusions.

Pg 29, ln 596: These statements are quite generalized, and should be refined. What variables should be added and what information content due they bring? What information is missing that could be provided by other sources and why are they not now included?

Pg 29, general: What are the limitations of the study? What is it's applicability?

———————————————

---

## Author Comment (AC1) · 26 Jan 2021

**Authors' response to interactive comments by Anonymous Referee #1**

January 26, 2021

Black text: Reviewer's comment

Blue text: Authors' response

**1  General Comments**

This manuscript describes a novel method for estimating snow water equivalent from snow depth and a variety of other variables, by using an ensemble of artificial neural networks (ANN). This type of machine learning model is becoming increasingly popular as computational power increases. Likewise, the estimation of snow water equivalent from remote sensing or other in-situ variables is in constant development due to the scarce availability of in-situ SWE measurements. The manuscript falls within two topics of current great scientific interest and within the scope of the journal. I first have to congratulate the authors for the huge amount of work that they have done in the analyses and the text. This is a follow-up study to Odry et al. (2020), who introduced the use of ANNs to estimate snow density from snow depth over Quebec, outperforming other regression models such as Jonas et al 2009 and Sturm et al 2010. Here, the authors develop and improve the initial model. As elements of novelty, the authors: (1) perform an in-depth analysis of model architecture, testing several options of model characteristics; (2) demonstrate that training a model for different snow classes improves model performance; and (3) use SWE as target variable instead of snow density, which also improves performance. In general, the text is well structured and well written. The introduction and literature review are lengthy but are useful for a non-specialised reader to understand the theory behind multilayer perceptrons (MLPs) and the model ensemble evaluation metrics. The experimental set up is very detailed, although it lacks some clarity in its structure. The results are extensive and support the conclusions reached by the authors. However, the authors should be more convincing about the scientific progress that their analyses provide, as compared to Odry et al. 2020. Moreover, the results are generally very descriptive but some more discussion

(or a discussion section) lacks. The results and conclusions are focused on the model performance improvement, but there is no discussion about the limitations of these type of models (e.g. the amount of data or computation time they require). There are also quite a few specific issues to be addressed to improve the quality, clarity, and reproducibility of the study. Overall, the work presented in this study is impressive, but the authors should address the issues that I outline here before I can recommend it for publication.

We would like to thank the reviewer for their extensive review including detailed comments and suggestions. It will strengthen the output of the study. Below we will address each specific comment and explain how we will incorporate them into the revised manuscript.

**2   Specific Comments**

1. " Lines 74-79: Here you should state more clearly what the novelty of this paper is compared to Odry et al. 2020. Before I finished reading the entire paper, I was not convinced that there would be enough novelty in this manuscript. Perhaps a good solution would be to add a few lines before this paragraph, summarising the key knowledge gaps that you are filling (testing model structures, target variables, including climate classifications,...), and why it is important to tackle these knowledge gaps. What is the aim of the paper besides just "improving a model"?

   We agree with the reviewer and will restructure the paragraph to include more specific information summarizing the results of the study. After presenting the key knowledge gaps, we will emphasize the main outputs of the study more precisely. The focus will be on using SWE instead of density as the target variable, Testing several options of ANN structural characteristics, including input uncertainty on snow depth and dividing the area into snow classes by using an individual ANN model for each of them to give a greater representation of the geophysical diversity of snow.

2. As estimated from line 309 and Figure 3(e), snow depth-SWE measurement sites have on average 100 records. This means that the model is mostly developed over independent and sparse measurements, rather than with time series. The influence of this is not discussed. Would MLPs benefit from training with long term time series? What would be the influence of including back-propagation loops in that case? Please discuss it. Testing it for a long-term dataset such as the SNOTEL dataset over the United Stated would be worth, but I acknowledge this would require an entire follow-up analysis.

   It is correct that snow depth-SWE measurement sites have got 100 records on average. However, in our model each record is treated individually, as the station ID is not an input in the MLP. Therefore, all data from all

sites belonging to a specific snow class are used to train and validate the model. The model is not trained on 100 records, but on much more (see Table 6 of the manuscript for the number of records per class). We will include the above clarification into the revised manuscript. Regarding continuous time series, the length of time series and why it is not central in this study, we would also like to add some precision. When using time series, no improvement is expected because of the nature of the time series, but because of the greater amount of data and probably more consistent measurements. To make use of a time series, one would need to use the snow depth of the previous days as an input of the MLP, which is not possible at the moment in Canada, as the Canadian snow survey includes only a few continuous time series, mainly in British Columbia due to snow pillows measuring SWE. Elsewhere, the data is not continuous in time. Our model was developed to use only what is available in operations in Canada and therefore only makes use of data available in real-time or near-real-time.

3. The explanatory variables in section 3.2 are identical to those in Odry et al. 2020, with the addition of snow density from ERA5, which turns out to be the least explanatory in Table 5. Why did not you include other variables suggested in Odry et al. 2020 such as wind and solar radiation?

Wind and solar radiation are not available in real time, but only through reanalysis (e.g. ERA5). Since the model is meant to be close to operational capabilities, we only want to include variables that are available in real-time. ERA5 snow density was primarily included as a test and the authors are pleased that this variable shows the lowest impact. After this result, no further variables from reanalysis where tested. We will provide more information on this in section 3.2 and a small discussion in section 4.1.4, mentioning that snow density will be excluded from operational use.

4. For variables of accumulated precipitation in the last n days, n is only tested for up to 10 days. In Table 1, three out of four variables show largest correlation for 10 days, which makes me suspect that the range tested is not big enough. Please test for a larger range. In case this becomes increasingly large, it could be that accumulated precipitation in the last n days and accumulated precipitation since the beginning of the winter are the "same explanatory variable".

The correlation was only calculated for a range of 1 to 10 days and no further investigation was done during the study. The two plots below show the Spearman's correlation for an extended range of 1 to 30 days. As mentioned by the reviewer, the correlation for solid and total precipitation increases with the number of days and therefore will lead to the same explanatory variable accumulated precipitation since the beginning of the winter. Note that also for snow density the Spearman's correlation does increase above 12 days and overreaches the first maximum at 3 days. We will mention this in the manuscript, but we would prefer to keep the input

variables as they are right now, as including input variables accounting for a longer time correlation would basically require recomputing everything from the start. This would unfortunately be very difficult for us at the moment, for time constraints reasons. However, the point of the reviewer is very valuable and will be considered in future studies.

[Figure]

[Figure]

5. The structure of Section 3 is somewhat confusing. I don't clearly under-
stand what is a "tested characteristic" and what is not. For instance, in-
put uncertainty and input variable selection are two "characteristics" that
are tested according to Table 2 and 3, but they have their own subsec-
tion (3.3 and 3.5 respectively), while the other tested characteristics (e.g.
optimization algorithm) are described in section 3.4. I suggest blending
section 3.5 and 3.2 together, and to include 3.3 and 3.6 in 3.4. Com-
bining Table 2 and Table 3 together would also help understand what is
being tested and what is not. Please either restructure the section, or
clearly justify the current one.The results in section 4.1 should then be
restructured according to the new structure of section 3.

We very much appreciate the suggestions, and we agree that the struc-
ture of the section can confuse the reader. We would like to propose a
different structure: 3.1 Data availability; 3.2 Input uncertainty (we would
like to break up the section. In 3.2 we explain only how the input uncer-
tainty is modelled); 3.3 Explanatory variables; 3.4 Tested characteristics
(we will take the suggestion of the reviewer and combine Table 2 and 3.
Furthermore, we will include the treatment of the input uncertainty within
the MLP, the input variable selection and the determination of number of
epochs and number of hidden neurons, because these are all character-
istics being tested in the study and being presented in the new combined
table.) In 3.4 Tested characteristics, we will work with subsections to get
a better structure in section 4.1.

6. Table 4: the combination (combo) of the best choice for each tested char-
acteristic provides even better performances for most evaluation metrics.
Why are all the characteristics tested one by one? Please test the com-
bined effect of characteristics or provide a clear explanation about why
this is not possible (computationally too expensive?) or not necessary.

The characteristics have been tested one by one to measure the effect
of each characteristic individually, following the Ceteris paribus principle.
After, that the combination of the characteristics were tested where im-
provement has been shown. Further testing of other combinations might
improve the system further. However, no large improvement is expected,
as the individual tests of the other characteristics showed no improve-
ment. We would like to mention that the testing is computationally expen-
sive because it runs over several numbers of epochs (2-200).

7. Line 472-473: What part of Section 4.1 are you referring to? It is not even
finished, 4.1.6 is still coming after this. I suggest to provide an extra table
(or a paragraph) with the final MLP architecture set-up, to avoid having to
jump section by section to gather all the final decisions.

Two different models are built. The first uses one MLP ensemble and is
applied to the entire area of Canada. This model is finalized in section
4.1.5. Section 4.1.6 finalizes the model using multiple MLP ensembles,
one for each snow class. As the structure of Section 3 will be changed,

we will also change the structure here (Section 4.1). We will also provide a table providing the final set up of both models.

8. "There is barely any discussion about computational cost of the final model architecture set-up, and about the number of epochs and hidden neurons. If applicable, I would like to see what the computational trade-off of the final model choices is (e.g.choosing for XXX is about 103 times slower than YYY, though I acknowledge this is CPU dependent)."

   We will provide the computational cost for the two final models (single MLP and multiple MLP) for the training on the training data set and the simulation of the testing data set.

9. Line 482-483: and the number of neurons decreases too. Please explain why.

   Snow classes with larger number of data points show higher variability in the records. This can be better represented by a network with more hidden neurons because complexity is increased. This information will be added to the revised version of the manuscript.

10. Lines 489-491: Why and what does it mean? This is the only reference to Figure 10 in the text. Please provide more information on the results shown in the figure. A discussion of the results is also necessary here, since there is no Discussion section.

    We can expand the discussion on Table 7. About Figure 10, we will clarify the caption, as we will do for all figures. We will clearly indicate which subfigures are reliability diagrams, which are rank histograms, etc. However, we are not sure what the reviewer means by "Why and what does it mean". The rank histogram is more reliable, because it is flatter with less outliers for the multiple MLP ensembles, presented by the first and last bar of the histogram. Furthermore, the reliability diagram shows a more reliable forecast when the points are closer to the identity line, which is the case for the multiple MLP ensembles. Both information are presented in section 2.4.3 and 2.4.4 when the evaluation metrics are introduced. We will provide a small explanation in the manuscript and refer to the sections 2.4.3. and 2.4.4. We hope that this will satisfy the reviewer's concerns.

11. Table 7: How are the values obtained for MMLP? Is it the median of the ensembles for each snow class? Please specify

    In both models (SMLP and MMLP), the median is taken for MAE, RMSE and MBE. We will introduce MAE, RMSE and MBE briefly in section 2.4, because this was asked by reviewer #2. In there, we will mention that the median is taken. For clarification, when simulating the test data set for each record, the snow class is determined and the associated MLP ensemble is taken in the multiple MLP ensembles model. This returns one ensemble for one record, as in the single MLP ensemble model.

12. Figure 10: top row: How is the "median simulation" obtained for a specific bin? How is the histogram of medians built? This is highly unclear, please specify. Also,why is there a small bin on the negative side? Does the model simulate negative SWE values? Middle row: Is this a reliability diagram? It can be guessed from the text but it is not specified anywhere else. Please provide a letter for all subpanels (a-f).

For clarity, we will add letters to all sub-panels and provide more information in the caption. Also, we will provide more information about the calculation of the median simulation in the caption. For each record, one ensemble is simulated from which we calculate the median. Further, both models simulate negative SWE values. For the single MLP ensemble the minimum of the simulation is $-36mm$ and for 0.6% of the records in testing data set the model simulates negative SWE values. For the multiple MLP ensembles model the minimum is $-42mm$ and the ratio of negative simulation is 0.3%. This information will be included in the revised manuscript, when Figure 10 is discussed in the text. As a side note, it is possible to get negative values for SWE, because the output layer is modelled by a linear function and therefore can output any value.

13. "Lines 495-496: Looking at figure 11 the values seem to be rather +/-18 and +/-16? Please provide the accurate values."

The exact values of the box are $[-17.1mm, 18.4mm]$ and for the whisker $[-77.6mm, 73.3mm]$ for the single MLP ensemble model. The exact values of the box are $[-15.8mm, 18.0mm]$ and for the whisker $[-68.8mm, 64.6mm]$ for the multiple MLP ensembles model. These values will be included in the revised version of the manuscript.

14. Line 552: In what figure/panel do you see the better accuracy for ephemeral snow? Please add this information.

We think that the reviewer is referring to line 502. The discussion refers to Figure 12(a) and will be included in the manuscript.

15. Line 507: Where is the deeper analysis of the ephemeral snow class shown? The text refers to a rank histogram for the snow class, but I don't see it. Same in line 520 referring to a rank histogram for mountain and maritime snow, where is it? Please show them.

We apologize that the rank histogram and reliability diagram are not presented in the manuscript for each snow class. We have tried to keep the manuscript as concise as possible. We would prefer not to include this large figure in the manuscript itself, and we would prefer to provide it as an additional material.

16. Figure 12: I believe the Y axis in panel (a) should be Skill Score (SS). Just as you did in Figure 13, please write "Skill Score" either on the axis label or in the caption, it is useful for the reader to be reminded what "SS" means. Same with RB, I had to read the text to find what it means

(Relative Bias). Why the SMLP is evaluated on RB and MMLP on MBE-SS?

*Our initial idea was to separate the SWE error metrics (MAE, RMSE, MBE) from the CRPS and the ignorance score, as it is written in the legend. Skill Score (SS) will be written on the y-axis for panel (a),(b) and (c), which is better understandable for the reader, as proposed by the reviewer. We will also include Relative Bias in either the legend or caption. Further, we would like to apologize for the mistake. In both cases the relative bias was taken. This will be corrected in the revised manuscript.*

17. Figure 14: Given the large amount of data, consider adding color to the scatter plots based on the density of points.

    *We thank you for the suggestion, we will present a color coded figure in the revised manuscript.*

18. Line 562: How many are "numerous"? It seems as if they are $< 10$? In that case, the MLP probably can't be trained for those high values of SWE, while the regression model is continuous also for high values. Please discuss the lack of training data for these high values.

    *In the testing data set there are 17 SWE measurement above $2500mm$. The training data set includes 18 SWE measurements above $2500mm$. Little training data in the higher range of SWE disables the MLP to estimate them correctly, because during training the model, it is adjusted such that the MSE over all data points is minimised. Therefore the model focuses on areas where the density of data point is the highest. We will provide a small discussion of this aspect in the revised manuscript.*

19. TITLE: The title of the manuscript is still too similar to Odry et al. 2020. Think of a title that would directly show the improvement/novelty with respect to Odry et al.2020. (e.g. use "testing ANN architectures", "snow class", "climatological variables" or"multiple ensembles")

    *We suggest changing the title to "Investigating ANN architectures and training to estimate SWE directly from snow depth".*

20. CAPTIONS: Throughout the manuscript, Figure and Table captions are just one line long. Please extend them. Make sure everything (acronyms, lines, points, etc.) that is shown in the Figure or Table is described on the legend, labels, or in the caption.

    *We believe we have already partially addressed this comment with our previous answers. We will provide more information for figures where it is needed in the revised manuscript.*

21. "DISCUSSION: Following many of my comments, provide discussion of results within the results section, or provide a separate discussion section before conclusions.I especially miss a discussion on limitations of the

model, applicability and transferability. How much data does the model need in order to be properly trained? You could for instance re-do the analyses but using only a random 10"

We would like to avoid a discussion section, because we think it is more convenient for the reader when the results are discussed when they are presented. We will extend our discussions in the result section. Furthermore, we will include some thoughts about limitations, applicability and transferability in the conclusion. Regarding the latter two, the used data set has got data records of all snow classes (except the ice snow class, for which we have too little records for a proper analysis) which shows the diversity of snow patterns within the study. Therefore, the model structure is expected to be applicable to other areas in the world. However, new training is advisable. Regarding limitation, both models show bad simulation results for high values, because the amount of training data is low. Further, the incorporation of climate change needs to be done manually or by taking data only from recent years. The amount of data needed to train the model properly cannot be answered universally. It depends on the variability of the data set. For instance, if an area shows many different snow patterns, more data is needed to get a satisfactory result. This also changes the number of epochs and number of neurons needed to get the best result. We always advise to check the model by a validation data set which is already required in many ANN libraries in Python.

**3   Technical corrections**

- Line 192: I'm guessing this is a typing error, otherwise why is DOYobs = 0 on 1st January? Shouldn't it be 123 provided 1st September is DOY = zero?

  It is not a typing error. $DOY_{obs}$ is $0$ in 1st of January and takes values from -122 till 243. This is consistent with the model proposed by Sturm et al. [2010]. However, we acknowledge our explanation was much too brief and therefore, it will be explained more precisely in the revised manuscript.

- All other technical correction comments are clear and will be included in the revised manuscript.

**References**

M. Sturm, B. Taras, G. E. Liston, C. Derksen, T. Jonas, and J. Lea. Estimating snow water equivalent using snow depth data and climate classes. *Journal of Hydrometeorology*, 11(6):1380–1394, 2010. doi: 10.1175/2010JHM1202.1.

---

## Author Comment (AC2) · 26 Jan 2021

**Authors' response to interactive comments by Anonymous Referee #2**

January 26, 2021

Black text: Reviewer's comment

Blue text: Authors' response

**1 General Comments**

This manuscript describes the application of machine learning techniques, specifically an ensemble of multilayer perceptrons, to estimate the hydrological variable Snow Water Equivalent (SWE). As described by the authors, SWE is a crucial variable which is difficult to directly measure at scale. The paper subject advances the application of ML techniques for SWE estimation by application of ensemble methods, discerning model applicability over climatic regions. and demonstrating advantages over empirical methods.

This research builds on the Ordy et al (2020) study by extending the geographical scope, using direct estimation of SWE and introducing snow classes in MLP training. There is sufficient novelty in this paper for publication, however it should first be strengthened in clear justification for decisions, interpretation of results and conclusion.

The manuscript leaves out some essential elements from Ordy et al, including descriptions of evaluation metrics and why they are selected. The addition of explanatory variables such as Snow Density from ERA5 lack explanation and background. Following strong results reporting sections, the discussion finding and conclusions of the manuscript require additional reflection on the limitations of the study and the context.

Overall, a strong effort worthy of publication on the basis of some revision and structural improvement. Some coherence is missing in the experimental design, in the inclusion of variables, the applicability of the study and the conclusions drawn from it. These require revision, hope the comments that follow can be of help.

We thank the reviewer for their comments and we appreciate the effort put in

the revision. We will take their comments into account to strengthen the output of the study. In the following, we address each specific comment and explain how we will incorporate them into the manuscript.

**2 Specific Comments**

1. Pg 1, ln 15: "Using a greater number of MLP parameters could lead to further improvements" It is somewhat self-evident that increased parameterization of an MLP model could potentially produce better results, can this statement be focused to the study specific outcomes?

   We will summarize the outcomes of the study more precisely. Specifically, we will focus on using SWE instead of density as the target variable, testing several options of ANN structural characteristics (e.g. optimization algorithm, activation function, parameter initialization, increasing the number of parameters) improves estimates of SWE, including input uncertainty on snow depth improves the model's performance and dividing the area into snow classes and using an individual ANN model for each of them gives a greater representation of the geophysical diversity of snow.

2. pg 2, 50: This description of the application of physics-based models for SWE estimation is a bit too simplistic here, given ERA5 snow density as used later as an explanatory variable. The iSnobal mentioned is a coupled energy and mass-balance model that requires a great deal of meteorological data derive accumulating snow density and in turn modelled SWE. Please provide some further description on the advanced requirements these approaches and limitations, beyond only computational cost

   Physical-based models like iSnobal take input variables (e.g. incoming longwave radiation, soil temperature, net solar radiation) which are not available in real time in Canada. Furthermore, physical-based models are, as Painter et al. [2016] mentioned, the logical choice for distributed SWE estimates. However, we aim for a conversion model based on data points which are sparely scattered in time and space and uses only variables available in real time. We will add these thoughts in the revised manuscript. Further, ERA5 snow density was included as a test of reanalysis data, but is not available in real-time. Further discussion is given in comment 13.

3. ln 54: Consider including Snauffer et al, 2018. https://doi.org/10.5194/tc-12-891-2018.

   We would like to thank the reviewer for mentioning related literature. We will include this article into the mentioned section.

4. pg 3, ln 76: The second hypotheses seems too broad. "in-depth testing". Please be more specific as to the methodology to be tested.

We do not want to change the hypothesis, because they have been determined before the study. However, we will be more specific with the outcomes of the study in this paragraph, in accordance with the answer to comment 1.

5. pg 3, ln 78: "The entire area of Canada." Is this an overreach given the the limited density of measurements across much of Canada? "Applicability in a broader context". Be more specific in the what this broader context is.

It was not our intention to overreach, but we wanted to mention that the data set is scattered sparsely and non-uniformly over the entire area of Canada, which is very large. Testing the "applicability in a broader context" means that the data set includes almost all the Sturm's snow classes (except the ice snow class, for which there is too little data to analyze), which gives the opportunity to test the model's applicability to multiple snow class zones. We will refine the explanations in the revised version of the manuscript accordingly.

6. pg 3, ln 78: This last sentence seems out of place to close the paragraph. Moving one sentence earlier would improve the paragraph.

In accordance with your comment, this sentence will be revised and probably linked together into one sentence.

7. pg 3, ln 94: Is MSE the definitive objective function for regression problems? Better likely to phrase as "commonly used"

Following the mathematical theory by Goodfellow et al. [2016], the MSE is derived from the the maximum likelihood estimator when dealing with regression models and therefore, the best choice. However, "commonly" can be included, since the determination of the objective function is ultimately the modeller's choice.

8. pg 5, ln 131: "The algorithms RMSProp and AdaDelta produce good results". Please elaborate in this statement, or tie in better with the following two sentences.

We are sorry that the formulation is misleading. The statement is indeed linked to the previous sentence. Schaul et al. [2014] compared stochastic gradient methods by testing them on small-scale problems, and concluded that the algorithms RMSProp and AdaDelta produced good results. This sentence will be modified in the revised manuscript to make it clearer.

9. pg 5, ln 155: Avoid starting sentences with "Because". This is a general comment also through the manuscript. Would recommend re-writing this initial sentence, breaking into sections.

We acknowledge the critiques and will account for it throughout the revised manuscript.

10. pg 6, ln 171: Can references be provided for some of these conclusions? The linkage of snow depth only to precipitation requires some basis. There are a lot of varied physical processes for snow accumulation between tundra and taiga eco-zones.

    In the revised version of the manuscript, we will provide further information on the specific characteristics of the different snow classes and refer to Sturm et al. [1995], who describe snow related characteristics for each snow class.

11. pg 8, ln 222: I don't follow the second sentence, though can be my ignorance. Consider a clearer explanation if including.

[Figure]

    $f$ is a probability density function (pdf). Pdfs output only values between $0$ and $1$. The $log_2$ function returns a value less than or equal $0$ for values between $0$ and $1$, as presented on the above plot. Subsequently, the negative of the $log_2$ function returns only values greater than or equal $0$. We will rephrase the sentence such that less mathematical formulas are included, but will not extend the explanation, since the theory section is already quite long.

12. pg 12, Figure 3e. It is possible to rescale the number of records for each site? It is not very descriptive. Is the maximum records up above 3000?

    One station within the data set has got 3203 records. Therefore, we would like to keep Figure 3(e) as it is to represent the entire data set. However, we acknowledge that the figure can be confusing. Therefore we will include the mean and maximum in the text when the other subpanels of the figure are discussed.

13. Ordy et al had recommended the inclusion of additional explanatory variables from meteorology, such as wind or solar radiation. This study has included ERA5 daily averaged snow density data. This output from a physically based model is included without description of its generation, assessment of the quality or the relevance or applicability of this data source. Although the variable is kept as least important for the conversion model, and minimal impact on the ignorance score, it is kept in complete assessment. The manuscript should include some rationale for the

inclusion of this model output, and why it was chosen. Are the assumptions in producing the snow density relevant? How does this data perform compared to available measurements?

Wind and solar radiation are not available in real time, but only through reanalysis (e.g. ERA5). Since the model is meant to be close to operations, we only want to include variables that are available in real-time. Snow density of ERA5 was included as a test and the authors are pleased that this variable shows the lowest impact. After this result, no further variables from reanalysis were tested. We will give more information in the section 3.2 of explanatory variable and a small discussion in section 4.1.4 of input variable selection and mention that is will be excluded for operational use.

14. Pg 15, Table 2, To clarify, on pg 4, ln 97 it is mentioned that modifying the order of the input data is recommended. Is that done in this study (Shuffling data before each epoch)?

Yes, this is part of the tested features in our study, as presented in Table 3, with the results in Table 4. However, this should become clearer when Section 3 is rearranged according to a comment by reviewer #1: 3.1 Data availability; 3.2 Input uncertainty (we would like to break up the section. In 3.2 we explain only how the input uncertainty is modelled); 3.3 Explanatory variables; 3.4 Tested characteristics. We will combine Table 2 and 3. We will also include the treatment of the input uncertainty within the MLP, the input variable selection and the determination of number of epochs and number of hidden neurons, because these are all characteristics being tested in the study and being presented in the new combined table. In 3.4 Tested characteristics, we will work with subsections to get a better structure in section 4.1.

15. Pg 17, Figure 5. MBE should be introduced before used. The use of error metrics is not entirely clear (MBE, MAE, RMSE) compared with clearly rational and description for other scores. Clearer rationale and explanation would help. Is there a reason the RMSE is shown compared to the objective function MSE? RMSE can be a more comparable error metric for SWE, but this is not explained.

We will introduce the metrics MAE, RMSE and MBE briefly in the Section 2.4 Model evaluation. RMSE is used here because it can be compared to MAE. RMSE penalizes large residuals compared to MAE. Also, RMSE has the same units as SWE and MAE whereas MSE does not and therefore, does not have any physical meaning.

16. Pg 20, line 449: This appears a notable and relevant finding (what explanatory variables are ultimately useful) that can be better articulated in study findings.

We will emphasize in the revised manuscript that five out of the six most important variables are coherent with the variable selection in Odry et al.

[2020]. However, we would also like to emphasis that the order of variables with scores lying close together can change, since the parameters are initialized randomly. Therefore, we can give a rough estimate of which variables are the most useful, but not an ultimate one. We will add this information to the revised version of the manuscript.

17. Pg 27, Figure 13: Consistency would be useful for interpretation between RB and MBE. Can see in the following paragraph why RB is substituted for MBE, but would like to see this graph included.

Unfortunately, we are not entirely sure we understand this comment correctly. We disagree with the suggestion of presenting the MBE over different snow classes. Different snow classes show different magnitudes of SWE, as presented in Figure 2. Therefore, showing the MBE disables a comparison between snow classes, as the MBE will necessarily be proportional to the magnitude of SWE for a given class. The MBE can only serve for a comparison between the models for each snow class individually.

18. Pg 27, line 555: The conclusions drawn in this section appear to have a relatively weak causal or testable links. Ranging from regression model structure, to physical processes to reference model performance, several comments seem quite speculative. For example, the tundra region has poor performance, but would be subject to may be similar topographic controls of the prairies. It would seem better to reflect on what information can truly be derived from these results, or at least address that there are many contributing factors that are not represented by this method.

We acknowledge the critiques and will delete or rephrase the speculative comments and focus on the actual output of the figure when discussing it in the text.

19. Pg 28: line 570: This opening sentence for the Conclusions section should be more descriptive and engaging in the content of the study

In accordance with comment 1 and 4, we will also change the first paragraph of the conclusion accordingly and take the reviewers suggestion into account.

20. Pg 29, ln 591: What is the additional geophysical information beyond snow class from Sturm et al.? If this refers the discretization by elevation class, this should be elaborated on in the rest of the document to include in conclusions.

The geophysical information is added by distributing the model into different snow classes. No further information was added. The reviewer's comment showed us that the formulation is misleading and we will delete it in the revised manuscript.

21. Pg 29, ln 596: These statements are quite generalized, and should be refined. What variables should be added and what information content

due they bring? What information is missing that could be provided by other sources and why are they not now included?

We would like to first test temperature and precipitation variables with longer time ranges. Further, we only want to look for variables which are available in real time or site specific. For instance, one could try topological variables, like slope or aspect. Also, one could test the model with more precise meteorological data on a limited number of sites. We will include these thoughts into the revised manuscript.

22. Pg 29, general: What are the limitations of the study? What is it's applicability?

We will include some thoughts about the limitations, applicability and transferability in the conclusion. Regarding the applicability and transferability, the used data set contains records for all snow classes (except of the ice snow class being to small for a proper analysis) which shows the diversity of snow patterns within the study. Therefore, the model structure is expected to be applicable to other areas in the world. However, new training is advisable. Regarding limitations, both models show bad simulation results for high values, because the amount of training data is low. With knowing this weakness of the model, this problem could easily be bypassed in an operational context with some sort of threshold above which only the background is used, for example. Furthermore, the incorporation of climate change needs to be done manually or by taking data only from recent years. The amount of data needed to train the model properly cannot be determined universally. It depends on the variability of the data set. For instance if an area shows many different snow patterns, more data is needed to get a satisfactory result. This also changes the number of epochs and number of neurons needed to get the best result. We always advice to check the model by a validation data set which is already required in many ANN libraries in Python.

**References**

I. Goodfellow, Y. Bengio, and A. Courville. *Deep Learning - Chapter 5 - 8*. MIT Press, 2016. `http://www.deeplearningbook.org`.

J. Odry, M. A. Boucher, P. Cantet, S. Lachance-Cloutier, R. Turcotte, and P. Y. St-Louis. Using artificial neural networks to estimate snow water equivalent from snow depth. *Canadian Water Resources Journal / Revue canadienne des ressources hydriques*, 0(0):1–17, 2020. doi: 10.1080/07011784.2020.1796817. URL `https://doi.org/10.1080/07011784.2020.1796817`.

T. H. Painter, D. F. Berisford, J. W. Boardman, K. J. Bormann, J. S. Deems, F. Gehrke, A. Hedrick, M. Joyce, R. Laidlaw, D. Marks, C. Mattmann,

B. McGurk, P. Ramirez, M. Richardson, S. M. Skiles, F. C. Seidel, and A. Winstral. The airborne snow observatory: Fusion of scanning lidar, imaging spectrometer, and physically-based modeling for mapping snow water equivalent and snow albedo. *Remote Sensing of Environment*, 184:139 − 152, 2016. doi: 10.1016/j.rse.2016.06.018.

T. Schaul, I. Antonoglou, and D. Silver. Unit Tests for Stochastic Optimization. *International Conference on Learning Representations (ICLR)*, 2014. https://arxiv.org/abs/1312.6055.

M. Sturm, J. Holmgren, and G. E. Liston. A Seasonal Snow Cover Classification System for Local to Global Applications. *Journal of Climate*, 8(5):1261–1283, May 1995. doi: 10.1175/1520-0442(1995)008¡1261:ASSCCS¿2.0.CO;2.

---

## Author Response (AR1)

**Authors' specific response regarding revision #1 to comments by Anonymous Referee #1**

March 9, 2021

Black text: Reviewer's comment

Blue text: Authors' response; The identifications of lines, figures and tables refer to the version with track changes.

**1 General Comments**

This manuscript describes a novel method for estimating snow water equivalent from snow depth and a variety of other variables, by using an ensemble of artificial neural networks (ANN). This type of machine learning model is becoming increasingly popular as computational power increases. Likewise, the estimation of snow water equivalent from remote sensing or other in-situ variables is in constant development due to the scarce availability of in-situ SWE measurements. The manuscript falls within two topics of current great scientific interest and within the scope of the journal. I first have to congratulate the authors for the huge amount of work that they have done in the analyses and the text. This is a follow-up study to Odry et al. (2020), who introduced the use of ANNs to estimate snow density from snow depth over Quebec, outperforming other regression models such as Jonas et al 2009 and Sturm et al 2010. Here, the authors develop and improve the initial model. As elements of novelty, the authors: (1) perform an in-depth analysis of model architecture, testing several options of model characteristics; (2) demonstrate that training a model for different snow classes improves model performance; and (3) use SWE as target variable instead of snow density, which also improves performance. In general, the text is well structured and well written. The introduction and literature review are lengthy but are useful for a non-specialised reader to understand the theory behind multilayer perceptrons (MLPs) and the model ensemble evaluation metrics. The experimental set up is very detailed, although it lacks some clarity in its structure. The results are extensive and support the conclusions reached by the authors. However, the authors should be more convincing about the scientific progress that their analyses provide, as compared to Odry et al. 2020.

Moreover, the results are generally very descriptive but some more discussion (or a discussion section) lacks. The results and conclusions are focused on the model performance improvement, but there is no discussion about the limitations of these type of models (e.g. the amount of data or computation time they require). There are also quite a few specific issues to be addressed to improve the quality, clarity, and reproducibility of the study. Overall, the work presented in this study is impressive, but the authors should address the issues that I outline here before I can recommend it for publication.

We would like to thank the reviewer for their extensive review including detailed comments and suggestions. It will strengthen the output of the study. Below we address each specific comment and explain how we incorporated them into the revised manuscript.

**2   Specific Comments**

1. ” Lines 74-79: Here you should state more clearly what the novelty of this paper is compared to Odry et al. 2020. Before I finished reading the entire paper, I was not convinced that there would be enough novelty in this manuscript. Perhaps a good solution would be to add a few lines before this paragraph, summarising the key knowledge gaps that you are filling (testing model structures, target variables, including climate classifications,...), and why it is important to tackle these knowledge gaps. What is the aim of the paper besides just "improving a model"?

   We agree with the reviewer and restructured the paragraph including more specific information, which summarizes the results of the study. In ln. 12-15, we added a more precise summary of the study in the abstract. The same information is presented in the introduction in ln. 84-92 in a broader fashion.

   It is correct that snow depth-SWE measurement sites have got roughly 100 records on average. However, in our model each record is treated individually, as the station ID is not an input in the MLP. Therefore, all data from all sites (belonging to a specific snow class for the multiple MLP ensembles model) are used to train and validate the model. The model is not trained on 100 records, but on much more (see Table 5 of the track change version for the number of records per class). We would also like to add some precision regarding continuous time series, the length of time series and why it is not central in this study. When using time series in our model, no improvement is expected because of the nature of the time series, but because of the greater amount of data and probably more consistent measurements. To make use of a time series, one would need to use the snow depth of the previous days as an input of the MLP, which is not possible at the moment in Canada, as the Canadian snow survey includes only a few continuous time series, mainly in British

Columbia due to snow pillows measuring SWE. Elsewhere, the data is not continuous in time. Our model was developed to use only what is available in operations in Canada and therefore only makes use of data available in real-time or near-real-time. We added information about the mean and maximum of the number of records per site in the text in ln. 353-354 and in the caption of Figure 3.

2. The explanatory variables in section 3.2 are identical to those in Odry et al. 2020, with the addition of snow density from ERA5, which turns out to be the least explanatory in Table 5. Why did not you include other variables suggested in Odry et al. 2020 such as wind and solar radiation?

Wind and solar radiation are not available in real time, but only through reanalysis (e.g. ERA5). Since the model is meant to be close to operational capabilities, we only want to include variables that are available in real-time. ERA5 snow density was primarily included as a test and the authors are pleased that this variable shows the lowest impact. After this result, no further variables from reanalysis where tested. We added in the section 3.2 "Explanatory variables" in ln. 370-371 that snow density from ERA5 is included as a test. In the result section in ln. 525-526, we give the information that snow density can be excluded for operational use.

3. For variables of accumulated precipitation in the last n days, n is only tested for up to 10 days. In Table 1, three out of four variables show largest correlation for 10 days, which makes me suspect that the range tested is not big enough. Please test for a larger range. In case this becomes increasingly large, it could be that accumulated precipitation in the last n days and accumulated precipitation since the beginning of the winter are the "same explanatory variable".

The correlation was only calculated for a range of 1 to 10 days and no further investigation was done during the study. The two plots below show the absolute Spearman's correlation for an extended range of 1 to 30 days. As mentioned by the reviewer, the correlation for solid and total precipitation increases with the number of days and therefore will lead to the same explanatory variable accumulated precipitation since the beginning of the winter. Note that also for snow density the Spearman's correlation increases after 12 days and overreaches the first maximum after 27 days. However, this shows that recent solid precipitation and accumulated precipitation over the entire winter season carry different information when looking at snow density. This is also pointed out by Odry et al. [2020] their Table 4. The impact of short term and long term variables with respect to SWE need to be reexamined. We mention in section 3.2 "Explanatory variables" in ln. 395-397 that we want to keep the information of short term variables. The above discussion is given in the conclusion in ln. 702-709.

[Figure]

[Figure]

4. The structure of Section 3 is somewhat confusing. I don't clearly understand what is a "tested characteristic" and what is not. For instance, input uncertainty and input variable selection are two "characteristics" that are tested according to Table 2 and 3, but they have their own subsection (3.3 and 3.5 respectively), while the other tested characteristics (e.g. optimization algorithm) are described in section 3.4. I suggest blending section 3.5 and 3.2 together, and to include 3.3 and 3.6 in 3.4. Combining Table 2 and Table 3 together would also help understand what is being tested and what is not. Please either restructure the section, or clearly justify the current one.The results in section 4.1 should then be restructured according to the new structure of section 3.

We very much appreciate the suggestions, and we agree that the structure of the section can confuse the reader. We restructured section 3 as follows: 3.1 Data availability; 3.2 Explanatory variables including input uncertainty; 3.3 Tested characteristics (we combined Table 2 and 3. Table

2 now shows all tested characteristics including input uncertainty within the MLP, the input variable selection and the determination of number of epochs and number of hidden neurons) This corresponds to ln. 399-446 and Table 2.

5. Table 4: the combination (combo) of the best choice for each tested characteristic provides even better performances for most evaluation metrics. Why are all the characteristics tested one by one? Please test the combined effect of characteristics or provide a clear explanation about why this is not possible (computationally too expensive?) or not necessary.

   The characteristics have been tested one by one to measure the effect of each characteristic individually, following the Ceteris paribus principle. After that the combination of the characteristics were tested where improvement has been shown. Additionnal testing of other combinations might improve the system further. However, no large improvement is expected, as the individual tests of the other characteristics showed no improvement. This information is given in ln. 425-427. We would also like to mention that the testing is computationally expensive because it runs over several numbers of epochs (mostly 2-200).

6. Line 472-473: What part of Section 4.1 are you referring to? It is not even finished, 4.1.6 is still coming after this. I suggest to provide an extra table (or a paragraph) with the final MLP architecture set-up, to avoid having to jump section by section to gather all the final decisions.

   For less confusion, we included the section 4.1.3 "Final setup of SMLP and MMLP" in ln. 564 and present the final setup in Table 6.

7. "There is barely any discussion about computational cost of the final model architecture set-up, and about the number of epochs and hidden neurons. If applicable, I would like to see what the computational trade-off of the final model choices is (e.g.choosing for XXX is about 103 times slower than YYY, though I acknowledge this is CPU dependent)."

   The discussion about computational cost for the two final models is presented in the section 4.1.3 "Final setup of SMLP and MMLP" in ln. 566-571.

8. Line 482-483: and the number of neurons decreases too. Please explain why.

   We included in ln. 561-563 that snow classes with smaller datasets show smaller variability in the records, which can be easily represented by a simpler network with less hidden neurons, because of the lower complexity of the problem.

9. Lines 489-491: Why and what does it mean? This is the only reference to Figure 10 in the text. Please provide more information on the results shown in the figure. A discussion of the results is also necessary here, since there is no Discussion section.

> We expanded the discussion on Table 7 and Figure 10 in ln. 574-582. Further, we tried to be more precise in the caption of Figure 10. We indicated which subfigures are reliability diagrams, which are rank histograms, etc. We included some discussion on the results of the rank histogram and the reliability diagram including a link to section 2.4.4 and 2.4.5, where the two evaluation metrics are introduced.

10. Table 7: How are the values obtained for MMLP? Is it the median of the ensembles for each snow class? Please specify

> In both models (SMLP and MMLP), the median is taken for MAE, RMSE and MBE. We introduced MAE, RMSE and MBE briefly in section 2.4.1 (ln.243-253), because this was asked by reviewer #2. In there, we mention that the median is taken. For clarification, when simulating the test data set for each record, the snow class is determined and the associated MLP ensemble is taken in the multiple MLP ensembles model. This returns one ensemble for one record, as in the single MLP ensemble model.

11. Figure 10: top row: How is the "median simulation" obtained for a specific bin? How is the histogram of medians built? This is highly unclear, please specify. Also,why is there a small bin on the negative side? Does the model simulate negative SWE values? Middle row: Is this a reliability diagram? It can be guessed from the text but it is not specified anywhere else. Please provide a letter for all subpanels (a-f).

> For clarity, we added letters to all sub-panels and provided more information in the caption in Figure 10. Also, we included more information about the calculation of the median simulation in the text in ln. 576. For each record, one ensemble is simulated from which we calculate the median. Further, both models simulate negative SWE values. For the single MLP ensemble the minimum of the simulation is $-36mm$ and for 0.6% of the records in the testing data set the model simulates negative SWE values. For the multiple MLP ensembles model the minimum is $-42mm$ and the ratio of negative simulation is 0.3%. This information is presented in ln. 577-579. As a side note, it is possible to get negative values for SWE, because the output layer is modelled by a linear function and therefore can output any value.

12. "Lines 495-496: Looking at figure 11 the values seem to be rather +/-18 and +/-16? Please provide the accurate values."

> The exact values of the box are $[-17.1mm, 18.4mm]$ and for the whisker $[-77.6mm, 73.3mm]$ for the single MLP ensemble model. The exact values of the box are $[-15.8mm, 18.0mm]$ and for the whisker $[-68.8mm, 64.6mm]$ for the multiple MLP ensembles model. This information is added in ln. 585-588.

13. Line 552: In what figure/panel do you see the better accuracy for ephemeral snow? Please add this information.

We think that the reviewer is referring to line 502. The discussion refers to Figure 12(a) and is included in ln. 594.

14. Line 507: Where is the deeper analysis of the ephemeral snow class shown? The text refers to a rank histogram for the snow class, but I don't see it. Same in line 520 referring to a rank histogram for mountain and maritime snow, where is it? Please show them.

We apologize that the rank histograms and are not presented in the manuscript for the *ephemeral, maritime* and *mountain* snow class. We have tried to keep the manuscript as concise as possible. We included Figure 13, showing the rank histogram for SMLP and MMLP within the *ephemeral* snow class. The corresponding discussion is given in ln. 599-601. We deleted the sentence which refers to the rank histogram for the *mountain* and *maritime* snow class, since no big difference is seen between the two snow classes, which already reflects the similar skill score.

15. Figure 12: I believe the Y axis in panel (a) should be Skill Score (SS). Just as you did in Figure 13, please write "Skill Score" either on the axis label or in the caption, it is useful for the reader to be reminded what "SS" means. Same with RB, I had to read the text to find what it means (Relative Bias). Why the SMLP is evaluated on RB and MMLP on MBE-SS?

Our initial idea was to separate the SWE error metrics (MAE, RMSE, MBE) from the CRPS and the ignorance score, as it is written in the legend. We changed the y-axis to Skill Score (SS) for panel (a),(b) and (c) in Figure 12, which is better understandable for the reader, as proposed by the reviewer. We also included "relative bias (RB)" in the caption. Further, we would like to apologize for the mistake. In both cases the relative bias was taken. This was corrected in the revised manuscript in Figure 12.

16. Figure 14: Given the large amount of data, consider adding color to the scatter plots based on the density of points.

Figure 15 was updated to a color coded scatter plot based on the density of points.

17. Line 562: How many are "numerous"? It seems as if they are $< 10$? In that case, the MLP probably can't be trained for those high values of SWE, while the regression model is continuous also for high values. Please discuss the lack of training data for these high values.

In the testing data set there are 17 SWE measurement above $2500mm$. The training data set includes 18 SWE measurements above $2500mm$. Little training data in the higher range of SWE disables the MLP to estimate them correctly, because during training the model, it is adjusted such that the MSE over all data points is minimised. Therefore the model

focuses on areas where the density of data point is the highest. This information is included in ln. 654-658.

18. TITLE: The title of the manuscript is still too similar to Odry et al. 2020. Think of a title that would directly show the improvement/novelty with respect to Odry et al.2020. (e.g. use "testing ANN architectures", "snow class", "climatological variables" or"multiple ensembles")

The title is changed to "Investigating ANN architectures and training to estimate SWE directly from snow depth".

19. CAPTIONS: Throughout the manuscript, Figure and Table captions are just one line long. Please extend them. Make sure everything (acronyms, lines, points, etc.) that is shown in the Figure or Table is described on the legend, labels, or in the caption.

We believe we have already partially addressed this comment with our previous answers. More information is provided in the captions of Figures 3, 5, 6, 7, 8, 9, 10 and 12.

20. "DISCUSSION: Following many of my comments, provide discussion of results within the results section, or provide a separate discussion section before conclusions.I especially miss a discussion on limitations of the model, applicability and transferability. How much data does the model need in order to be properly trained? You could for instance re-do the analyses but using only a random 10"

We would like to avoid a discussion section, because we think it is more convenient for the reader when the results are discussed when they are presented. We extended our discussions in the result section in correspondence with the above comments. Furthermore, some thoughts about limitations, applicability and transferability are included in the conclusion. Regarding the latter two, the used data set has got data records of all snow classes (except the ice snow class, for which we have too little records for a proper analysis) which shows the diversity of snow patterns within the study. Therefore, the model structure is expected to be applicable to other areas in the world. However, new training is advisable. This information is included in ln. 696-699. Regarding limitation, both models show bad simulation results for high values, because the amount of training data is low. Furthermore, the model is not predictive and especially cannot account for the effect of climate change. The amount of data needed to train the model properly cannot be answered universally. It depends on the variability of the data set. For instance, if an area shows many different snow patterns, more data is needed to get a satisfactory result. This also changes the number of epochs and number of neurons needed to get the best result. We always advise to check the model by a validation data set, which is already required in many ANN libraries in Python. This information is included in ln. 713-720.

**3   Technical corrections**

- Line 75: Please replace "verify" for "test", otherwise it seems that the hypotheses have been customised based on your results.

  This is changed in ln. 86.

- Line 192: I'm guessing this is a typing error, otherwise why is DOYobs = 0 on 1st January? Shouldn't it be 123 provided 1st September is DOY = zero?

  It is not a typing error. $DOY_{obs}$ is $0$ in 1st of January and takes values from -122 till 243. This is consistent with the model proposed by Sturm et al. [2010]. However, we acknowledge our explanation was much too brief and therefore, it is explained more precisely in ln. 213-214.

- Line 457: Figure 8a-e Figure 8: panel g should be f, and h should be g. Also, I suggest making a 4x2 (or 2x4) panel figure, with the only empty panel filled with the legend. It would reduce white space.

  We welcome the suggestion of the reviewer and changed the layout and corrected the letter identifications of the subplots in Figure 8.

- Line 526: "Second, [...]" but where is first? Rephrase accordingly.

  We rephrased this in ln. 618.

  item Line 588: Remove "the remainder of".

  We removed "the remainder of" in ln. 689.

**References**

J. Odry, M. A. Boucher, P. Cantet, S. Lachance-Cloutier, R. Turcotte, and P. Y. St-Louis. Using artificial neural networks to estimate snow water equivalent from snow depth. *Canadian Water Resources Journal / Revue canadienne des ressources hydriques*, 0(0):1–17, 2020. doi: 10.1080/07011784.2020.1796817. URL `https://doi.org/10.1080/07011784.2020.1796817`.

M. Sturm, B. Taras, G. E. Liston, C. Derksen, T. Jonas, and J. Lea. Estimating snow water equivalent using snow depth data and climate classes. *Journal of Hydrometeorology*, 11(6):1380–1394, 2010. doi: 10.1175/2010JHM1202.1.

**Authors' specific response regarding revision #1 to comments by Anonymous Referee #2**

March 9, 2021

Black text: Reviewer's comment

Blue text: Authors' response; The identification of lines, figures and tables refer to the version with track changes.

**1 General Comments**

This manuscript describes the application of machine learning techniques, specifically an ensemble of multilayer perceptrons, to estimate the hydrological variable Snow Water Equivalent (SWE). As described by the authors, SWE is a crucial variable which is difficult to directly measure at scale. The paper subject advances the application of ML techniques for SWE estimation by application of ensemble methods, discerning model applicability over climatic regions. and demonstrating advantages over empirical methods.

This research builds on the Ordy et al (2020) study by extending the geographical scope, using direct estimation of SWE and introducing snow classes in MLP training. There is sufficient novelty in this paper for publication, however it should first be strengthened in clear justification for decisions, interpretation of results and conclusion.

The manuscript leaves out some essential elements from Ordy et al, including descriptions of evaluation metrics and why they are selected. The addition of explanatory variables such as Snow Density from ERA5 lack explanation and background. Following strong results reporting sections, the discussion finding and conclusions of the manuscript require additional reflection on the limitations of the study and the context.

Overall, a strong effort worthy of publication on the basis of some revision and structural improvement. Some coherence is missing in the experimental design, in the inclusion of variables, the applicability of the study and the conclusions drawn from it. These require revision, hope the comments that follow can be of help.

We thank the reviewer for their comments and we appreciate the effort put in the revision. We tried to take their comments into account, which we believe strengthened the output of the study. In the following, we address each specific comment and explain how we incorporated them into the manuscript.

**2 Specific Comments**

1. Pg 1, ln 15: "Using a greater number of MLP parameters could lead to further improvements" It is somewhat self-evident that increased parameterization of an MLP model could potentially produce better results, can this statement be focused to the study specific outcomes?

   We summarized the outcomes of the study more precisely. Specifically, we first focus on using SWE instead of density as the target variable. Second, testing several options of ANN structural characteristics (e.g. optimization algorithm, activation function, parameter initialization, increasing the number of parameters) improves estimates of SWE. Third, including input uncertainty on snow depth improves the model's performance and dividing the area into snow classes. Fourth, using an individual ANN model for each of them gives a greater representation of the geophysical diversity of snow. This information is included in ln. 12-18.

2. pg 2, 50: This description of the application of physics-based models for SWE estimation is a bit too simplistic here, given ERA5 snow density as used later as an explanatory variable. The iSnobal mentioned is a coupled energy and mass-balance model that requires a great deal of meteorological data derive accumulating snow density and in turn modelled SWE. Please provide some further description on the advanced requirements these approaches and limitations, beyond only computational cost

   Physics-based models like iSnobal take input variables (e.g. incoming longwave radiation, soil temperature, net solar radiation) which are not available in real time in Canada. Furthermore, physics-based models are, as Painter et al. [2016] mentioned, the logical choice for distributed SWE estimates. However, we aim for a conversion model based on data points which are sparely scattered in time and space and uses only variables available in real time. These thoughts were added in ln. 54-59. Further, ERA5 snow density was included as a test of reanalysis data, but is not available in real-time. Further discussion is given in comment 13.

3. ln 54: Consider including Snauffer et al, 2018. https://doi.org/10.5194/tc-12-891-2018.

   We would like to thank the reviewer for mentioning related literature. We included this article into the introduction in ln. 60-62.

4. pg 3, ln 76: The second hypotheses seems too broad. "in-depth testing". Please be more specific as to the methodology to be tested.

We do not want to change the hypothesis, because they have been determined before the study. However, we tried to be more specific with the outcomes of the study in ln. 87-92.

5. pg 3, ln 78: "The entire area of Canada." Is this an overreach given the the limited density of measurements across much of Canada? "Applicability in a broader context". Be more specific in the what this broader context is.

It was not our intention to overreach, but we wanted to mention that the data set is scattered sparsely and non-uniformly over the entire area of Canada, which is very large. Testing the "applicability in a broader context" means that the data set includes almost all the Sturm's snow classes (except the ice snow class, for which there is too little data to analyze), which gives the opportunity to test the model's applicability to multiple snow class zones. We refined the explanations in the revised version of the manuscript accordingly in ln. 92-95.

6. pg 3, ln 78: This last sentence seems out of place to close the paragraph. Moving one sentence earlier would improve the paragraph.

This sentence was deleted and the information was reformulated in accordance with the previous comment, referring to ln. 92-95.

7. pg 3, ln 94: Is MSE the definitive objective function for regression problems? Better likely to phrase as "commonly used"

Following the mathematical theory by Goodfellow et al. [2016], the MSE is derived from the the maximum likelihood estimator when dealing with regression models and therefore, the best choice. However, "commonly" was included in ln. 111, since the determination of the objective function is ultimately the modeller's choice.

8. pg 5, ln 131: "The algorithms RMSProp and AdaDelta produce good results". Please elaborate in this statement, or tie in better with the following two sentences.

The statement is linked to the previous sentence. Schaul et al. [2014] compared stochastic gradient methods by testing them on small-scale problems, and concluded that the algorithms RMSProp and AdaDelta produced good results. This sentence is tied to the previous in ln. 149.

9. pg 5, ln 155: Avoid starting sentences with "Because". This is a general comment also through the manuscript. Would recommend re-writing this initial sentence, breaking into sections.

We acknowledge the critiques and accounted for it throughout the revised manuscript in ln. 144-145, 173-176, 184-185, 324-325, 332-334, 358-359.

10. pg 6, ln 171: Can references be provided for some of these conclusions? The linkage of snow depth only to precipitation requires some basis. There are a lot of varied physical processes for snow accumulation between tundra and taiga eco-zones.

   We provided further information on the specific characteristics of the different snow classes and refer to Sturm et al. [1995], who describe snow related characteristics for each snow class in ln.193-195.

11. pg 8, ln 222: I don't follow the second sentence, though can be my ignorance. Consider a clearer explanation if including.

[Figure]

   $f$ is a probability density function (pdf). Pdfs output only values between $0$ and $1$. The $log_2$ function returns a value less than or equal $0$ for values between $0$ and $1$, as presented on the above plot. Subsequently, the negative of the $log_2$ function returns only values greater than or equal $0$. We rephrased the sentence such that less mathematical formulas are included in ln. 257-256.

12. pg 12, Figure 3e. It is possible to rescale the number of records for each site? It is not very descriptive. Is the maximum records up above 3000?

   One site within the data set has got 3203 records. Therefore, we would like to keep Figure 3(e) as it is to represent the entire data set. However, we acknowledge that the figure can be confusing. Therefore we included the mean and maximum of the number of records per site in the text in ln 353-355 and in the caption of Figure 3.

13. Ordy et al had recommended the inclusion of additional explanatory variables from meteorology, such as wind or solar radiation. This study has included ERA5 daily averaged snow density data. This output from a physically based model is included without description of its generation, assessment of the quality or the relevance or applicability of this data source. Although the variable is kept as least important for the conversion model, and minimal impact on the ignorance score, it is kept in complete assessment. The manuscript should include some rationale for the inclusion of this model output, and why it was chosen. Are the assumptions in producing the snow density relevant? How does this data perform compared to available measurements?

Wind and solar radiation are not available in real time, but only through re-analysis (e.g. ERA5). Since the model is meant to be close to operation, we only want to include variables that are available in real-time. Snow density of ERA5 was included as a test and the authors are pleased that this variable shows the lowest impact. After this result, no further variables from reanalysis were tested. We added in the section 3.2 "Explanatory variables" in ln. 370-371 that snow density from ERA5 is included as a test. In the result section in ln. 525-526, we give the information that snow density can be excluded for operational use.

14. Pg 15, Table 2, To clarify, on pg 4, ln 97 it is mentioned that modifying the order of the input data is recommended. Is that done in this study (Shuffling data before each epoch)?

Yes, this is part of the tested features in our study. The Section 3 is re-arranged according to a comment by reviewer #1 to the following: 3.1 Data availability; 3.2 Explanatory variables including input uncertainty; 3.3 Tested characteristics (we combined Table 2 and 3. Table 2 shows now all tested characteristics including input uncertainty within the MLP, the input variable selection and the determination of number of epochs and number of hidden neurons) This corresponds to ln. 399-446. Now, Table 2 shows the *Reference* setup and all "Options" being tested throughout the study.

15. Pg 17, Figure 5. MBE should be introduced before used. The use of error metrics is not entirely clear (MBE, MAE, RMSE) compared with clearly rational and description for other scores. Clearer rationale and explanation would help. Is there a reason the RMSE is shown compared to the objective function MSE? RMSE can be a more comparable error metric for SWE, but this is not explained.

We introduced the metrics MAE, RMSE and MBE briefly in the Section 2.4.1 in ln. 243-253. RMSE is used here because it can be compared to MAE. RMSE penalizes large residuals compared to MAE. Also, RMSE has the same units as SWE and MAE whereas MSE does not and therefore, does not have any physical meaning.

16. Pg 20, line 449: This appears a notable and relevant finding (what explanatory variables are ultimately useful) that can be better articulated in study findings.

We emphasized in ln. 513-515 that five out of the six most important variables are coherent with the variable selection in Odry et al. [2020]. However, we also emphasized that the order of variables with scores lying close together can change, since the parameters are initialized randomly. Therefore, we can give a rough estimate of which variables are the most useful, but not an ultimate one.

17. Pg 27, Figure 13: Consistency would be useful for interpretation between RB and MBE. Can see in the following paragraph why RB is substituted for MBE, but would like to see this graph included.

Unfortunately, we are not entirely sure if we understand this comment correctly. We disagree with the suggestion of presenting the MBE over different snow classes. Different snow classes show different magnitudes of SWE, as presented in Figure 2. Therefore, showing the MBE disables a comparison between snow classes, as the MBE will necessarily be proportional to the magnitude of SWE for a given class. The MBE can only serve for a comparison between the models for each snow class individually.

18. Pg 27, line 555: The conclusions drawn in this section appear to have a relatively weak causal or testable links. Ranging from regression model structure, to physical processes to reference model performance, several comments seem quite speculative. For example, the tundra region has poor performance, but would be subject to may be similar topographic controls of the prairies. It would seem better to reflect on what information can truly be derived from these results, or at least address that there are many contributing factors that are not represented by this method.

We acknowledge the critiques and deleted or rephrased the speculative comments with focus on the actual output of the Figure 14 when discussing it in the text. The changes are done in ln. 647-650.

19. Pg 28: line 570: This opening sentence for the Conclusions section should be more descriptive and engaging in the content of the study

In accordance with comment 1 and 4, we also changed the first paragraph of the conclusion in ln. 686-688 accordingly and take the reviewers suggestion into account.

20. Pg 29, ln 591: What is the additional geophysical information beyond snow class from Sturm et al.? If this refers the discretization by elevation class, this should be elaborated on in the rest of the document to include in conclusions.

The geophysical information is added by distributing the model into different snow classes. No further information was added. The reviewer's comment showed us that the formulation is misleading and we deleted it in ln. 692.

21. Pg 29, ln 596: These statements are quite generalized, and should be refined. What variables should be added and what information content due they bring? What information is missing that could be provided by other sources and why are they not now included?

After proposing SWE as the new target variable in this study, short term and long term variables regarding precipitation with respect to SWE need to be analyzed. This also indicates Table 1. Furthermore, we only want

to look for variables which are available in real time or site specific. E.g. topological variables like the slope and aspect of measurement site can be used. These thought and further discussion are included in the manuscript in ln. 702-712.

22. Pg 29, general: What are the limitations of the study? What is it's applicability?

Some thoughts about limitations, applicability and transferability are included in the conclusion. Regarding the latter two, the dataset used in this study has data records of all snow classes (except the ice snow class, for which we have too little records for a proper analysis) which shows the diversity of snow patterns within the study. Therefore, the model structure is expected to be applicable to other areas in the world. However, new training is advisable. This information is included in ln. 696-699. Regarding limitation, both models show bad simulation results for high values, because the amount of training data is low. Furthermore, the model is not predictive and especially cannot account for the effect of climate change. The amount of data needed to train the model properly cannot be answered universally. It depends on the variability of the data set. For instance, if an area shows many different snow patterns, more data is needed to get a satisfactory result. This also changes the number of epochs and number of neurons needed to get the best result. We always advise to check the model by a validation data set which is already required in many ANN libraries in Python. This information is included in ln. 713-720.

**References**

I. Goodfellow, Y. Bengio, and A. Courville. *Deep Learning - Chapter 5 - 8*. MIT Press, 2016. `http://www.deeplearningbook.org`.

J. Odry, M. A. Boucher, P. Cantet, S. Lachance-Cloutier, R. Turcotte, and P. Y. St-Louis. Using artificial neural networks to estimate snow water equivalent from snow depth. *Canadian Water Resources Journal / Revue canadienne des ressources hydriques*, 0(0):1–17, 2020. doi: 10.1080/07011784.2020.1796817. URL `https://doi.org/10.1080/07011784.2020.1796817`.

T. H. Painter, D. F. Berisford, J. W. Boardman, K. J. Bormann, J. S. Deems, F. Gehrke, A. Hedrick, M. Joyce, R. Laidlaw, D. Marks, C. Mattmann, B. McGurk, P. Ramirez, M. Richardson, S. M. Skiles, F. C. Seidel, and A. Winstral. The airborne snow observatory: Fusion of scanning lidar, imaging spectrometer, and physically-based modeling for mapping snow water equivalent and snow albedo. *Remote Sensing of Environment*, 184:139 – 152, 2016. doi: 10.1016/j.rse.2016.06.018.

T. Schaul, I. Antonoglou, and D. Silver. Unit Tests for Stochastic Optimization. *International Conference on Learning Representations (ICLR)*, 2014. `https://arxiv.org/abs/1312.6055`.

M. Sturm, J. Holmgren, and G. E. Liston. A Seasonal Snow Cover Classification System for Local to Global Applications. *Journal of Climate*, 8(5):1261–1283, May 1995. doi: 10.1175/1520-0442(1995)008¡1261:ASSCCS¿2.0.CO;2.

---

## Referee Report (RR1)

**Review of revised manuscript for "Using an ensemble of artificial neural networks to convert snow depth to snow water equivalent over Canada" – Konstantin Franz Fotios Ntokas, Jean Odry, Marie-Amelie Boucher and Camille Garnaud.**

**Now: "Investigating ANN architectures and training to estimate SWE directly from snow depth"**

**General Comments**

Dear authors and editor,

I am pleased to see the authors have taken my comments into consideration, and I am happy with their reply for all comments that I do not re-discuss below. The manuscript has clearly improved its structure and readability, and I (again) want to state that the method presented here to estimate SWE is a good contribution to scientific progress that deserves to be published. In fact, most of my comments do not relate to the validity or robustness of the methods, but to the contextualisation of the approach, its applicability, and some statements made by the authors. In my opinion there are still some issues that need to be seriously addressed before publication. Most of them relate to the answers you have given to my previous report, but a couple new minor issues have arisen from re-reviewing the text. Line numbers refer to the **track changes version** of the manuscript.

1) I am still not convinced about how you contextualise your study and your aims. You have to convince the reader that your manuscript/method provides a relevant improvement with respect to Odry et al 2020. It is not just about writing "these are the knowledge gaps", as you do in line 84, but also why those knowledge gaps must be tackled. The knowledge gaps must logically arise from and be linked to the introduction (especially the paragraph before describing the results of Odry et al 2020). Something like: "This is a follow-up study [..]. While they did XXX, they did not consider PPP, so here we further test YYY and ZZZ, which is important because NNN. We hypothesize that (hypothesis 1) and (hypothesis 2). Furthermore, we aim to…". Also, line 92 is confusing when you write "We also take the opportunity to…". It reads as if that is a new aim and another third dataset, but to me it sounds like a repetition from the previous sentence. Please think about this carefully from a reader's perspective.

2) Thank you for your explanation about why not to use snow depth time series. However, your argument that you want to use data that is only available in near-real-time has made me realise that there is something inconsistent in your aims and method, and the following issues are linked to each other. You don't have and don't need real-time snow depth data, but you do need real-time precipitation data. Therefore, in what circumstances is the model going to be really applicable for operational use? I am guessing that you will only be able to use it at sites where there is real-time meteorological data available, and then a single snow depth measurement is provided at some point in time. From my ignorance about operational use over Canada, is this a realistic application? If so, this must be stated more clearly somewhere. This links to my comments on the aims of the study and the method. If not, you should reconsider what the aim of this approach to estimate SWE is (I think it can be very valuable for several applications, but this should be clearer in the text).

3) Related to the previous comment, the ANN is trained, validated and tested with ERA5 meteorological data, but these are not available in real time (you discard snow density from ERA5 because it is not available in real time). Therefore, I am assuming in real-time operational use, only in-situ meteorological station data will be used. Therefore, we don't really know if the ANN will perform well for the real-time operational application. You apply a lapse rate for temperature, but precipitation can also vary a lot between ERA5 and point

locations. The model will be trained from dynamics and features of reanalysis data, which can differ from station data. Again, if you want to keep the real-time operational use as one main aim for your method, then an independent validation should be provided with station data. Are there meteorological data available for some of your snow survey locations? If so, you should provide an additional validation for real-time use.

4) I have realised that it is not right to say that your method estimates SWE directly from snow depth (which is now even in the title, so I do not think it is accurate). You also need meteorological data. Given that, your method might be more comparable to a temperature index model, than to the simple regression models that you compare it with, which need only snow depth and simple geographical data (elevation, region, day of the year…). For instance, a recently published paper estimates SWE directly from snow heights ( https://doi.org/10.5194/hess-25-1165-2021 ), but they really only need snow height and its temporal change. I think it should be stated, especially in the introduction and conclusions, why you decide to compare your ANN with simple regression models (Jonas, Sturm), given that your method requires more data, and it is then not surprising that it performs better. This should also be a limitation of the method, but even if I suggested that you write more about limitations, you only added that the ANN does not perform well for the very high and very low values of snow density. The amount of data required does not only mean "how large the data is" but also the type of data. In that sense, your method requires more data than other simple regression models.

5) Regarding Table 1, and your new statement in line 706-707. I agree that information on the short term time scales is relevant, especially due to fresh snow density effects. However, I think the justification to include short term accumulated precipitation comes rather from the effect on snow density. Even if it is not the target variable, the effects might be still "hidden" in SWE. Given that correlation between SWE and "n days precip." increases, I find the choice of 10 days arbitrary, because it is not justified by the data. What is the effect on the score of explanatory variables if you choose (or add) n=3 or n=5 instead?

6) The new structure of section 3 and 4 is great, as well as Table 2 and 3! Much clearer and logical now.

7) After more thoughts on Figure 10, I think the histograms on (a) and (b) provide little information. It is hard to compare simulations vs observation, but also (a) vs. (b). I think a scatter plot would be a lot more informative. Since you do not mention outliers here (or very high values) you could cut the x and y axis to 2000mm. I know the scatter is already shown in Figure 15, but here it would provide zoomed in information. Similarly, the scatter plots in Figure 15 should be cut to 4000mm, or even 3000, as long as it is stated in the text that some outliers (probably 0.0001%) are outside the figure limits. Further, why is the origin of Figure 15 not at zero-zero? Also, include "colour shows scatter density" in the caption. Similar applies to Figure 3a,b,c,e, the x axis should be cut to where the bins are not visible anymore.

8) Line 676-678. I agree, but then it might be worth adding the Odry 2020 configuration in Table 6, for comparison.

9) Finally, I suggest that the github repository to reproduce the study is a little clearer. It took me a long while to understand the logical order of the codes, and what the folders are and where they come from. A clearer README file explaining the workflow (in addition to the figure in "OO.Overview.pdf" would be highly appreciated.

**Other technical corrections:**

- Table 5 is not referenced in the text anymore. You "lost" it when crossing it in line 555.

- I like your clarification (to me) about how the MLP ensemble works: "For clarification, when simulating the test data set for each record, the snow class is determined and the associated MLP ensemble is taken in the multiple MLP ensembles model. This returns one ensemble for one record, as in the single MLP ensemble model." It should be included in the text.
- Line 367: I think it is better "From snow depth, snow density, total precipitation, and temperature, we obtain the following explanatory variables."
- Line 425: Rephrase, it is hard to read. Maybe "the characteristics shown in Table 2 (first six rows in Options column), are tested…" ?
- Line 445: Should be "Sec. 4.1.3 and 4.1.4."
- Line 566: Swap order of MMLP and SSMLP, for consistency with the rest of the manuscript where SMLP is shown first.
- Line 567: Take (not takes).
- Line 575: "All performance metrics are smaller, except MBE."
- Line 599: Should be "ephemeral snow class in Fig. 13".
- Line 689: Perhaps reiterate here what the large gain in reliability is based on (what figure or metric).
- Line 703. A dot instead of comma after "analysed".
- Line 709: "such as" instead of "e.g.".

---

## Author Response (AR2)

**Authors' specific response regarding revision #2 to comments by Anonymous Referee #1**

April 22, 2021

Black text: Reviewer's comment

Blue text: Authors' response; The identification of lines, figures and tables refer to the version with track changes.

**1 General Comments**

I am pleased to see the authors have taken my comments into consideration, and I am happy with their reply for all comments that I do not re-discuss below. The manuscript has clearly improved its structure and readability, and I (again) want to state that the method presented here to estimate SWE is a good contribution to scientific progress that deserves to be published. In fact, most of my comments do not relate to the validity or robustness of the methods, but to the contextualisation of the approach, its applicability, and some statements made by the authors. In my opinion there are still some issues that need to be seriously addressed before publication. Most of them relate to the answers you have given to my previous report, but a couple new minor issues have arisen from rereviewing the text. Line numbers refer to the track changes version of the manuscript

The authors would like to thank the referee #1. The review will surely improve our study further. In the following, we address each specific comment and explain how we incorporated them into the manuscript.

**2 Specific Comments**

1. I am still not convinced about how you contextualise your study and your aims. You have to convince the reader that your manuscript/method provides a relevant improvement with respect to Odry et al 2020. It is not just about writing "these are the knowledge gaps", as you do in line 84, but also why those knowledge gaps must be tackled. The knowledge

gap must logically arise from and be linked to the introduction (especially the paragraph before describing the results of Odry et al 2020). Something like: "This is a follow-up study [..]. While they did XXX, they did not consider PPP, so here we further test YYY and ZZZ, which is important because NNN. We hypothesize that (hypothesis 1) and (hypothesis 2). Furthermore, we aim to...". Also, line 92 is confusing when you write "We also take the opportunity to...". It reads as if that is a new aim and another third dataset, but to me it sounds like a repetition from the previous sentence. Please think about this carefully from a reader's perspective.

We thank the referee for the critical thinking from a reader's perspective. We rewrote the paragraph in line 80-100, focused on the gaps in Odry et al. [2020] and tried to logically derive the aims and goals of the current study.

2. Thank you for your explanation about why not to use snow depth time series. However, your argument that you want to use data that is only available in near-real-time has made me realise that there is something inconsistent in your aims and method, and the following issues are linked to each other. You don't have and don't need real-time snow depth data, but you do need real-time precipitation data. Therefore, in what circumstances is the model going to be really applicable for operational use? I am guessing that you will only be able to use it at sites where there is real-time meteorological data available, and then a single snow depth measurement is provided at some point in time. From my ignorance about operational use over Canada, is this a realistic application? If so, this must be stated more clearly somewhere. This links to my comments on the aims of the study and the method. If not, you should reconsider what the aim of this approach to estimate SWE is (I think it can be very valuable for several applications, but this should be clearer in the text).

We address comment 2 and 3 together after comment 3.

3. ln 54: Related to the previous comment, the ANN is trained, validated and tested with ERA5 meteorological data, but these are not available in real time (you discard snow density from ERA5 because it is not available in real time). Therefore, I am assuming in real-time operational use, only in-situ meteorological station data will be used. Therefore, we don't really know if the ANN will perform well for the real-time operational application. You apply a lapse rate for temperature, but precipitation can also vary a lot between ERA5 and point4) locations. The model will be trained from dynamics and features of reanalysis data, which can differ from station data. Again, if you want to keep the real-time operational use as one main aim for your method, then an independent validation should be provided with station data. Are there meteorological data available for some of your snow survey locations? If so, you should provide an additional validation for real-time use.

We would like to clarify the concept behind the study and address comments 2 and 3 together.

The ANN ensemble is trained on manual snow surveys where snow depth and SWE are both available. Furthermore, the meteorological data for training comes from a reanalysis, in this study from ERA 5. The main goal of the study is to test the applicability of ANN ensembles to the conversion problem. With this respect, it appeared coherent to test it using the longest historical record possible, using meteorological data that is consistent in time (i.e. reanalysis) so that the differences in performances can by mainly attributed to the ANN model itself. The actual application in real time will be a second step that is more specific to the used atmospheric and hydrological forecasting system. In operational use, the recently available Regional Deterministic Reforecast System (RDRS; Gasset et al. [2021, in revision] in revision) would be used. Note that this product was just released and was not available in time for this study. RDRS has similar dynamics and physics as the operational Global Environmental Multiscale Model (GEM) used operationally at Environment and Climate Change Canada. The historical records are much shorter for this kind of data (snow depth and SWE) and also it is not consistent for the whole period. Some testing would be required before RDRS can actually be used, but it will be more specific to the forecasting system.

When simulating SWE, in-situ snow depth measurements could be taken (e.g. the sonic sensors provided by Meteorological Service of Canada and ECCC [2020]). As meteorological data, one could use an operational "nowcast" from an atmospheric model that include a land data assimilation system such as the Canadian Land Data Assimilation System (CaLDAS; Carrera et al. [01 Jun. 2015]). CalDAS is forced by real-time precipitation analyses from the Canadian Precipitation Analysis (CaPA; Fortin et al. [2015]), which combines simulated background precipitation fields with observed data (in situ and radars). Furthermore, the proposed method can be applied onto assimilated snow depth data in CaLDAS. Currently in CaLDAS, only snow depth data is assimilated and subsequently converted to SWE using the simulated density to initialize the land surface scheme. The proposed method would allow for two important upgrades: First, it would allow to assimilate snow depth data (converted to SWE) as well as SWE data, thus increasing the quantity of assimilated observations, and second, it would avoid using the simulated density, which is very hard to simulate accurately. In the introduction (line 80-81), we now explain that we built a model to estimate SWE from in-situ snow depth measurements and several indicators derived from gridded meteorological time series. The information about operational use as stated in this response is given in the conclusion, line 689-701.

4. I have realised that it is not right to say that your method estimates SWE directly from snow depth (which is now even in the title, so I do not think it is accurate). You also need meteorological data. Given that, your method

might be more comparable to a temperature index model, than to the simple regression models that you compare it with, which need only snow depth and simple geographical data (elevation, region, day of the year...). For instance, a recently published paper estimates SWE directly from snow heights (https://doi.org/10.5194/hess-25-1165-2021 ), but they really only need snow height and its temporal change. I think it should be stated, especially in the introduction and conclusions, why you decide to compare your ANN with simple regression models (Jonas, Sturm), given that your method requires more data, and it is then not surprising that it performs better. This should also be a limitation of the method, but even if I suggested that you write more about limitations, you only added that the ANN does not perform well for the very high and very low values of snow density. The amount of data required does not only mean "how large the data is" but also the type of data. In that sense, your method requires more data than other simple regression models.

We wanted to emphasis in the title the new finding of the current study that SWE is the direct output of the ANN ensemble model. We think that our model is probably an intermediate between snow modeling and regression model. Neural networks are in fact multivariate regressions. So they are comparable to regression models, they are just a more complex regression. The model uses meteorological time series, but derives certain indicators from them and cannot be used in the same way as a degree-day model because it does not dynamically accumulates nor melts snow. The only purpose of this model is to convert snow depth to SWE, and it was intended like that (not as a replacement of a snow accumulation and melt model). Actually, the ANN only sees the indicators, not the time series. Theoretically, it would be possible to build a regression using the exact same inputs used by the proposed ANN, but it would probably be more complicated to deal with the data (inter-correlation, normality, non-linearity,...). ANN is known to be more powerful to identify non-linear relationships among variables than regression. However, for less confusion, we deleted 'directly' in the title and reformulated it in line 84. Further, we added some information in the conclusion about model limitation regarding data requirements in line 681-683.

5. Regarding Table 1, and your new statement in line 706-707. I agree that information on the short term time scales is relevant, especially due to fresh snow density effects. However, I think the justification to include short term accumulated precipitation comes rather from the effect on snow density. Even if it is not the target variable, the effects might be still "hidden" in SWE. Given that correlation between SWE and "n days precip." increases, I find the choice of 10 days arbitrary, because it is not justified by the data. What is the effect on the score of explanatory variables if you choose (or add) n=3 or n=5 instead?

We agree with the referee that the choice of 10 days is taken arbitrarily and further investigation of the explanatory variables needs to be done to

cover short term effects with respect to SWE as the new target variable. This information has been added to the manuscript in revision #1 in line 667-674. This study took the same input variables as the previous study Odry et al. [2020] (except snow density from ERA5) and changed the output variable to identify the effect of that change. We do not believe that a random calculation of n=3 or n=5 gives us a lot of information at this point, and thus will not perform them.

6. The new structure of section 3 and 4 is great, as well as Table 2 and 3! Much clearer and logical now.

    We would like to thank the referee for his helpful suggestion and are happy that it made the manuscript clearer.

7. After more thoughts on Figure 10, I think the histograms on (a) and (b) provide little information. It is hard to compare simulations vs observation, but also (a) vs. (b). I think a scatter plot would be a lot more informative. Since you do not mention outliers here (or very high values) you could cut the x and y axis to 2000mm. I know the scatter is already shown in Figure 15, but here it would provide zoomed in information. Similarly, the scatter plots in Figure 15 should be cut to 4000mm, or even 3000, as long as it is stated in the text that some outliers (probably 0.0001%) are outside the figure limits. Further, why is the origin of Figure 15 not at zero-zero? Also, include "colour shows scatter density" in the caption. Similar applies to Figure 3a,b,c,e, the x axis should be cut to where the bins are not visible anymore.

    We agree that the histograms in Fig. 10a and b provide little information and we would like to take up the suggestion of the referee. We show a zoom in of the scatter plot with adjusted axes in Fig. 10a and b and give some information in the caption. Similarly, we adjusted the axes in Fig. 15 and state the portion of outliers not shown in the caption and line 619-620. Furthermore, we updated the x-axis of Figure 3a,b,c and e and give the information about non-shown outlier in the caption and line 356-358.

8. Line 676-678. I agree, but then it might be worth adding the Odry 2020 configuration in Table 6, for comparison.

    In Table 6, we added the configuration of the ANN ensemble of the previous study, proposed by Odry et al. [2020]: The table is linked to the conclusion statement in line 644.

9. Finally, I suggest that the github repository to reproduce the study is a little clearer. It took me a long while to understand the logical order of the codes, and what the folders are and where they come from. A clearer README file explaining the workflow (in addition to the figure in "OO.Overview.pdf" would be highly appreciated.

    We refined the README file in the github repository to improve the guidance through the code structure.

**3  Technical Corrections**

All the technical corrections were made, and we indicate the lines where they appear in the revised version with track change.

- Table 5 is not referenced in the text anymore. You "lost" it when crossing it in line 555. line 533-534

- I like your clarification (to me) about how the MLP ensemble works: "For clarification, when simulating the test data set for each record, the snow class is determined and the associated MLP ensemble is taken in the multiple MLP ensembles model. This returns one ensemble for one record, as in the single MLP ensemble model." It should be included in the text. line 426-428

- Line 367: I think it is better "From snow depth, snow density, total precipitation, and temperature, we obtain the following explanatory variables." line 371-372

- Line 425: Rephrase, it is hard to read. Maybe "the characteristics shown in Table 2 (first six rows in Options column), are tested..." ? line 417-418

- Line 445: Should be "Sec. 4.1.3 and 4.1.4." line 434; note that this was a problem with latexdiff in the previous revision. Thus, it does not appear in blue in the current revision.

- Line 566: Swap order of MMLP and SSMLP, for consistency with the rest of the manuscript where SMLP is shown first. line 539-540

- Line 567: Take (not takes). line 540

- Line 575: "All performance metrics are smaller, except MBE." line 547

- Line 599: Should be "ephemeral snow class in Fig. 13". line 569

- Line 689: Perhaps reiterate here what the large gain in reliability is based on (what figure or metric). line 656

- Line 703. A dot instead of comma after "analysed". line 668

- Line 709: "such as" instead of "e.g.". line 674

**References**

M. L. Carrera, S. Bélair, and B. Bilodeau. The canadian land data assimilation system (caldas): Description and synthetic evaluation study. *Journal of Hydrometeorology*, 16(3):1293 – 1314, 01 Jun. 2015. doi: 10.1175/JHM-D-14-0089.1. URL https://journals.ametsoc.org/view/journals/hydr/16/3/jhm-d-14-0089$_1$.$xml$.

V. Fortin, G. Roy, N. Donaldson, and A. Mahidjiba. Assimilation of radar quantitative precipitation estimations in the canadian precipitation analysis (capa). *Journal of Hydrology*, 531:296–307, 2015. ISSN 0022-1694. doi: https://doi.org/10.1016/j.jhydrol.2015.08.003. URL `https://www.sciencedirect.com/science/article/pii/S0022169415005624`. Hydrologic Applications of Weather Radar.

N. Gasset, V. Fortin, M. Dimitrijevic, M. Carrera, B. Bilodeau, R. Muncaster, E. Gaborit, G. Roy, N. Pentcheva, M. Bulat, X. Wang, R. Pavlovic, F. Lespinas, and D. Khedhaouiria. A 10 km north american precipitation and land surface reanalysis based on the gem atmospheric model. *Hydrology and Earth System Sciences Discussions*, 2021:1–50, 2021, in revision. doi: 10.5194/hess-2021-41. URL `https://hess.copernicus.org/preprints/hess-2021-41/`.

Meteorological Service of Canada and ECCC. Historical climate data, 2020. date of access: Oct 2019, `https://dd.weather.gc.ca/climate/observations/`.

J. Odry, M. A. Boucher, P. Cantet, S. Lachance-Cloutier, R. Turcotte, and P. Y. St-Louis. Using artificial neural networks to estimate snow water equivalent from snow depth. *Canadian Water Resources Journal / Revue canadienne des ressources hydriques*, 0(0):1–17, 2020. doi: 10.1080/07011784.2020.1796817. URL `https://doi.org/10.1080/07011784.2020.1796817`.